# Semi-supervised Active Linear Regression

**Fnu Devvrit**[*]
Department of Computer Science
University of Texas at Austin
devvrit@cs.utexas.edu

**Nived Rajaraman**[*]
Department of Electrical Engineering and Computer Sciences
University of California, Berkeley
nived.rajaraman@berkeley.edu

**Pranjal Awasthi**
Google Research & Department of Computer Science
Rutgers University
pranjal.awasthi@rutgers.edu

## Abstract

Labeled data often comes at a high cost as it may require recruiting human labelers or running costly' experiments. At the same time, in many practical scenarios, one already has access to a partially labeled, potentially biased dataset that can help with the learning task at hand. Motivated by such settings, we formally initiate a study of *semi-supervised active learning* through the frame of linear regression. Here, the learner has access to a dataset $X \in \mathbb{R}^{(n_{\mathrm{un}}+n_{\mathrm{lab}}) \times d}$ composed of $n_{\mathrm{un}}$ unlabeled examples that a learner can actively query, and $n_{\mathrm{lab}}$ examples labeled a priori. Denoting the true labels by $Y \in \mathbb{R}^{n_{\mathrm{un}}+n_{\mathrm{lab}}}$, the learner's objective is to find $\widehat{\beta} \in \mathbb{R}^d$ such that,

$$\|X\widehat{\beta} - Y\|_2^2 \leq (1 + \epsilon) \min_{\beta \in \mathbb{R}^d} \|X\beta - Y\|_2^2 \tag{1}$$

while querying the labels of as few unlabeled points as possible. In this paper, we introduce an instance dependent parameter called the reduced rank, denoted $\mathrm{R}_X$, and propose an efficient algorithm with query complexity $O(\mathrm{R}_X/\epsilon)$. This result directly implies improved upper bounds for two important special cases: $(i)$ active ridge regression, and $(ii)$ active kernel ridge regression, where the reduced-rank equates to the *statistical dimension*, $\mathsf{sd}_\lambda$ and *effective dimension*, $d_\lambda$ of the problem respectively, where $\lambda \geq 0$ denotes the regularization parameter. Finally, we introduce a distributional version of the problem as a special case of the agnostic formulation we consider earlier; here, for every $X$, we prove a matching instance-wise lower bound of $\Omega(\mathrm{R}_X/\epsilon)$ on the query complexity of any algorithm.

## 1 Introduction

Classification and regression are the cornerstone of machine learning. Often these tasks are carried out in a supervised learning manner - a learner collects many examples and their labels, and then runs an optimization algorithm to minimize a loss function and learn a model. However, the process

---

[*]equal contribution

36th Conference on Neural Information Processing Systems (NeurIPS 2022).

of collecting labeled data comes at a cost, for instance when a human labeler is required, or if an expensive experiment needs to be run for the same. Motivated by such scenarios, two popular approaches have been proposed to mitigate these issues in practice: Active Learning [24] and Semi-Supervised Learning [30]. In Active Learning, the learner carries out the task by *adaptively* querying the labels of a small subset of characteristic data points. On the other hand, semi-supervised learning is motivated by the fact that learners often have access to massive amounts of unlabeled data in addition to some labeled data, and algorithms leverage both to carry out the learning task. Active learning and Semi-supervised learning have found numerous applications in areas such as text classification [26], visual recognition [17] and foreground-background segmentation [15].

In practice, learners often have access to smaller labeled datasets and larger unlabeled datasets, chosen points of which can be submitted for manual labeling. This motivates the *semi-supervised active regression* (SSAR) formulation - here, the learner is provided an a priori labeled data $X_{\text{lab}} \in \mathbb{R}^{n_{\text{lab}} \times d}$, as well as an unlabeled dataset $X_{\text{un}} \in \mathbb{R}^{n_{\text{un}} \times d}$, points of which can be iteratively submitted to an external agent to be labeled at unit cost. In the linear regression framework, the objective of the learner is to then minimize the squared-loss $\|X\beta - Y\|_2^2$ while making as few label queries (of points in $X_{\text{un}}$) as possible. Here $X$ is the overall dataset, $X_{\text{lab}} \cup X_{\text{un}}$ represented as a matrix in $\mathbb{R}^{(n_{\text{un}} + n_{\text{lab}}) \times d}$, while $Y \in \mathbb{R}^{n_{\text{un}} + n_{\text{lab}}}$ denotes the corresponding labels. The learner seeks to find a regression function $\widehat{\beta} \in \mathbb{R}^d$ such that,

$$\|X\widehat{\beta} - Y\|_2^2 \leq (1 + \epsilon) \min_{\beta \in \mathbb{R}^d} \|X\beta - Y\|_2^2. \tag{2}$$

Likewise, the $\epsilon$-query complexity of an algorithm is defined as the number of unlabeled points in $X_{\text{un}}$ that must be queried in order to return a $\widehat{\beta}$ such that eq. (2) is satisfied with constant probability.

In the special case when no labeled data is provided (i.e. $X_{\text{lab}}$ is empty), SSAR reduces to active linear regression, which has been studied in several prior works [9, 5, 7, 8, 6].

SSAR includes several other important problems as special cases. For some $\lambda > 0$, consider the setting where the labeled dataset $X_{\text{lab}}$ is the set of standard basis vectors scaled by a factor of $\sqrt{\lambda}$, with corresponding labels as 0. The loss of a learner $\widehat{\beta}$ on this dataset exactly corresponds to the ridge regression loss on the unlabeled dataset, $\|X_{\text{un}}\widehat{\beta} - Y_{\text{un}}\|_2^2 + \lambda\|\beta\|_2^2$, where $Y_{\text{un}}$ are the labels of points in $X_{\text{un}}$. Thus, *active ridge regression* is captured as a special case of SSAR.

On the other hand, it turns out that *active kernel ridge regression* is also captured as a special case of SSAR. Here, data points lie in a reproducing kernel hilbert space (RKHS) [21, 25, 20] that is potentially infinite dimensional, but the learner has access to a kernel representation of the $N$ data points. Denoting the kernel matrix by $K \in \mathbb{R}^{N \times N}$ and the labels of the points as $Y \in \mathbb{R}^N$, the objective in kernel ridge regression is to minimize the loss $\|K\beta - Y'\|_2^2 + \lambda\|\beta\|_K^2$. This can be viewed as a special case of the SSAR framework by choosing the labeled dataset as $X_{\text{lab}} = \sqrt{\lambda}\sqrt{K} \in \mathbb{R}^{N \times N}$ with each point labeled by 0, and the unlabeled dataset $X_{\text{un}} = K \in \mathbb{R}^{N \times N}$ with labels $Y' \in \mathbb{R}^N$.

In this work, the main question we ask is,

### Can we design algorithms for SSAR with instance-wise optimal query complexities?

Our first main contribution is to propose an instance dependent parameter called the *reduced rank*, denoted $\mathsf{R}_X$, that can be used to upper bound the worst-case optimal query complexity of SSAR. Intuitively, $\mathsf{R}_X$ measures the relative importance of the points at which labels are known, $X_{\text{lab}}$, in the context of the overall dataset. Formally, the reduced rank is defined as:

$$\mathsf{R}_X = \mathsf{Tr}\left(\left(X_{\text{un}}^T X_{\text{un}} + X_{\text{lab}}^T X_{\text{lab}}\right)^{-1} X_{\text{un}}^T X_{\text{un}}\right) \tag{3}$$

We propose an efficient polynomial time algorithm for SSAR with $\epsilon$-query complexity bounded by $O(\mathsf{R}_X/\epsilon)$. In the special cases of active ridge regression and kernel ridge regression, $\mathsf{R}_X$ is shown to be the equal to the *statistical dimension* $\mathsf{sd}_\lambda$ [3] and the *effective dimension* $d_\lambda$ [28] of the problem respectively, which directly implies query complexity guarantees for the two special cases. These parameters are formally defined in Section 3.

**Optimal algorithms for active ridge regression.** The recent work of [6] studies the problem of active regression and following a long line of work (see [9]) provides a near optimal (up to constant factors) algorithm for active regression querying labels of at most $O(d/\epsilon)$ points. The algorithm of [6]

can be used for active ridge regression too, where $\lambda \neq 0$, resulting in a query complexity guarantee of $O(d/\epsilon)$. However, the dependence on $d$ is not ideal in this setting and an instance-dependent parameter of the data known as *statistical dimensions* $\mathsf{sd}_\lambda$ is the right measure of the complexity of the problem. As a consequence of our general algorithm and guarantee, we obtain an algorithm for active ridge regression with query complexity upper bounded by $O(\mathsf{sd}_\lambda/\epsilon)$, which we show is in fact optimal in the worst case. As a result we resolve complexity of ridge regression in the active setting.

**Improved guarantees for active kernel ridge regression.** In the context of kernel ridge regression, the work of [1] analyzed kernel leverage score sampling (and in general Nyström approximation based methods), showing that these sampling approaches query $O(d_\lambda \log(N))$ labels to find a constant factor approximation to the optimal loss. In contrast, our algorithm when applied in the kernel ridge regression setting achieves a query complexity guarantee of $O(d_\lambda/\epsilon)$ bearing no dependence on the number of datapoints, $N$, which is often large in practice.

**Techniques.** Our algorithm is based on a novel variant of the Randomized BSS *spectral sparsification* algorithm of [16, 4], that returns a small weighted subset of the dataset, queries its labels, and subsequently carries out weighted least-squares regression on this dataset. The key technical contribution of this work is to construct a sampling algorithm which is guaranteed to query the labels of only a limited number of unlabeled points. We discuss the algorithm and its guarantees in more detail in Section 4.

The rest of the paper is organized as follows. We cover related work in Section 1.1 and introduce relevant preliminaries in Section 2. We motivate and present the new parameter $\mathrm{R}_X$ in Section 3, and present our algorithm for SSAR and a proof sketch in Section 4. In Section 5 we introduce the distributional SSAR problem, and show a matching instance-dependent lower bound for the problem.

## 1.1 Related Work

Several prior works use active learning to augment semi-supervised learning based approaches [12, 29, 10, 22, 23]. The key in several of these works is to use active learning methods to decide which labels to sample, and then use semi-supervised learning to finally learn the model.

In the context of active learning for linear regression, [9, 27] analyze the leverage score sampling algorithm showing a sample complexity upper bound of $O(d \log(d)/\epsilon)$. Subsequently for the case of $\epsilon \geq d + 1$, [7] shows that volume sampling achieves an $O(d)$ sample complexity. Later, [8] shows that rescaled volume sampling matches the sample complexity of $O(d \log(d)/\epsilon)$ achieved by leverage score sampling. Finally, [6] shows that a spectral sparsification based approach using the Randomized BSS algorithm [16] achieves the optimal sample complexity of $O(d/\epsilon)$.

In the context of kernel ridge regression, the works of [11, 28, 19] study the problem of reducing the runtime and number of kernel evaluations to approximately solve the problem. These results show that the number of kernel evaluations can be made to scale only linearly with the effective dimension of the problem, $d_\lambda(K)$ up to log factors. Recently, [1] prove a statistical guarantee showing that any optimal Nystrom approximation based approach samples at most $d_\lambda(K) \log(n)$ labels to find a constant factor approximation to the kernel regression loss, where $n$ is the dimension of the kernel.

A closely related problem to active learning is the coreset (and weak coreset) problem [2, 18], where the objective is to find a low-memory data structure that can be used to approximately reconstruct the loss function which naïvely requires storing all the training examples to compute. [13] study the weak coreset problem for ridge regression and propose an algorithm that returns a weak coreset of $O(\mathsf{sd}_\lambda/\epsilon)$ points which can approximately recover the optimal loss. A drawback of their algorithm is that the labels of all the points in the dataset are required, which may come at a very high cost in practice. As we discuss in Section 4, our algorithm also returns a weighted subset of $O(\mathsf{sd}_\lambda/\epsilon)$ points in $\mathbb{R}^d$ and a set of $d$ additional weights that can be used to reconstruct a $(1 + \epsilon)$-approximation for the original ridge regression instance. In terms of memory requirement, this matches the result of [13]; however it suffices for our algorithm to query *at most $O(\mathsf{sd}_\lambda/\epsilon)$ labels*.

## 2 Preliminaries

In this section, we formally define Semi-supervised Active Regression (SSAR) under squared-loss in the linear setting. The learner is provided datasets $X_{\mathrm{un}} \in \mathbb{R}^{n_{\mathrm{un}} \times d}$ (comprised of $n_{\mathrm{un}}$ points in $\mathbb{R}^d$) and

| Setting | Upper bounds | Lower bounds |
|---|---|---|
| Agnostic SSAR[†] | $\frac{R_X \cdot \log(d)}{\epsilon}$ (LSS[⋆]: Theorem 1) 
 $\frac{R_X}{\epsilon}$ (ASURA: Theorem 3 + 4) | - |
| Distributional SSAR[††] | $\frac{R_X}{\epsilon}$ (ASURA: Corollary 1) | $\frac{R_X}{\epsilon}$ (Theorem 5) |

Table 1: Table of guarantees for SSAR

[†] The agnostic SSAR problem assumes no distribution on the input, and assumes no label noise.
[††] The distributional SSAR problem (Section 5) assumes that the input follows a known distribution and queried labels are corrupted by Gaussian noise.
[⋆] LSS stands for Leverage Score Sampling.

| Setting | Upper bounds | Lower bounds |
|---|---|---|
| Active Ridge Regression | $\frac{d \cdot \log(d)}{\epsilon}$ (LSS[⋆] [9, 13]) 
 $\frac{d}{\epsilon}$ (Linear Sparsification [6]) 
 $\frac{\mathsf{sd}_\lambda}{\epsilon}$ (ASURA: Theorem 3 + 4) | $\frac{\mathsf{sd}_\lambda}{\epsilon}$ (Theorem 5) |
| Active Kernel Ridge Regression | $\frac{d_\lambda \cdot \log(n)}{\epsilon}$ (Nystrom Approximation [1]) 
 $\frac{d_\lambda}{\epsilon}$ (ASURA: Corollary 1) | $\frac{d_\lambda}{\epsilon}$ (Theorem 5) |

Table 2: Table of guarantees for special cases of SSAR
[⋆] LSS stands for Leverage Score Sampling.

$X_{\mathrm{lab}} \in \mathbb{R}^{n_{\mathrm{lab}} \times d}$. The labels of the points are respectively $Y_{\mathrm{un}} \in \mathbb{R}^{n_{\mathrm{un}}}$ and $Y_{\mathrm{lab}} \in \mathbb{R}^{n_{\mathrm{lab}}}$. Define,

$$X = \begin{bmatrix} X_{\mathrm{un}} \\ X_{\mathrm{lab}} \end{bmatrix}, \quad Y = \begin{bmatrix} Y_{\mathrm{un}} \\ Y_{\mathrm{lab}} \end{bmatrix} \tag{4}$$

In the agnostic setting, the labels of the points are not assumed to be generated from an unknown underlying linear function and can be arbitrary. We study the problem in the overconstrained setting with $n_{\mathrm{un}} + n_{\mathrm{lab}} \geq d$. Furthermore, in the active setting, it is typically the case that $n_{\mathrm{un}} \gg d$. When clear from the context, we use $X$, $X_{\mathrm{lab}}$ and $X_{\mathrm{un}}$ to represent the datasets in set notation.

We assume that the learner is a priori provided $Y_{\mathrm{lab}}$ and hence knows the labels of all points in $X_{\mathrm{lab}}$, but the labels of points in $X_{\mathrm{un}}$ are not known. However, the learner can probe an oracle to return the label of any queried point in $X_{\mathrm{un}}$ and incurs a unit cost for each label returned. The learner's objective is to return a linear function $\widehat{\beta} \in \mathbb{R}^d$ which approximately minimizes the squared $\ell_2$ loss,

$$\|X\widehat{\beta} - Y\|_2^2 \leq (1+\epsilon)\|X\beta^* - Y\|_2^2, \quad \text{where,} \quad \beta^* \triangleq \underset{\beta \in \mathbb{R}^d}{\arg\min} \|X\beta - Y\|_2^2. \tag{5}$$

SSAR has the following two important problems as subcases.

**Active ridge regression:** In the active ridge regression problem, the learner is provided an unlabeled dataset $X_{\mathrm{un}} \in \mathbb{R}^{n \times d}$ of $n$ points. The learner has access to an oracle which can be actively queried to label points in $X_{\mathrm{un}}$. Representing the labels by $Y_{\mathrm{un}} \in \mathbb{R}^n$, the objective of the learner is to minimize $\mathcal{L}_\lambda(\beta) \triangleq \|X_{\mathrm{un}}\beta - Y_{\mathrm{un}}\|_2^2 + \lambda\|\beta\|_2^2$. Defining $X = \begin{bmatrix} X_{\mathrm{un}} \\ \sqrt{\lambda}I \end{bmatrix} \in \mathbb{R}^{(n+d) \times d}$ and $Y = \begin{bmatrix} Y_{\mathrm{un}} \\ 0 \end{bmatrix} \in \mathbb{R}^{n+d}$, observe that $\|X\beta - Y\|_2^2 = \|X_{\mathrm{un}}\beta - Y_{\mathrm{un}}\|_2^2 + \lambda\|\beta\|_2^2$. Thus, active ridge regression is a special case of the SSAR objective in eq. (5).

**Active kernel ridge regression:** In the kernel ridge regression problem, the learner operates in a reproducing kernel Hilbert space $\mathcal{H}$ and is provided an unlabeled dataset $X$ of $n$ points with (implicit) feature representations $\Phi(x)$ for $x \in X$. The objective is to learn a linear function $F(x)$, in the feature representations of points in $X$ that minimizes the squared $\ell_2$ norm with regularization. Namely,

$\sum_{i=1}^{n}(F(x_i) - Y_i)^2 + \lambda\|F\|_{\mathcal{H}}^2$. By the Representer Theorem [14], it turns out the optimal regression function $F^*(\cdot)$ can be expressed as $\sum_{i=1}^{n}\beta_i\kappa(x_i, \cdot)$ where $\kappa(\cdot, \cdot)$ is the kernel function. The kernel ridge regression objective can be expressed in terms of the kernel matrix $K \triangleq [\kappa(x_i, x_j)]_{i,j=1}^{n} \in \mathbb{R}^{n \times n}$. In other words, denoting the labels by $Y_{\text{un}} \in \mathbb{R}^n$ and $\|\beta\|_K^2 = \beta^T K\beta$, the learner's objective is to minimize over $\beta \in \mathbb{R}^n$,

$$\|K\beta - Y_{\text{un}}\|_2^2 + \lambda\|\beta\|_K^2 \tag{6}$$

while minimizing the number of label queries of points in $X_{\text{un}}$. The objective can be represented as,

$$\min_{\beta \in \mathbb{R}^n} \left\| \begin{bmatrix} K \\ \sqrt{\lambda}\sqrt{K} \end{bmatrix} \beta - \begin{bmatrix} Y_{\text{un}} \\ 0 \end{bmatrix} \right\|_2^2 \tag{7}$$

where $Z = \sqrt{K}$ is defined as the unique solution to $K = Z^T Z$. Yet again, it is a special case of the linear regression objective in eq. (5) with $X_{\text{un}} = K$, $X_{\text{lab}} = \sqrt{K}$ and $Y_{\text{lab}} = 0$.

## 3 Motivating the reduced rank $\mathbf{R}_X$

A natural algorithm for SSAR is to use the popular approach of leverage score sampling [9, 13]. This approach uses the entire dataset $X$ (both labeled and unlabeled) to construct a diagonal weight matrix $W_S$: here $S$ represents the support of the diagonal and $\mathbb{E}[|S|] \triangleq m = O(d\log(d)/\epsilon)$. The learner subsequently solves the problem: $\widehat{\beta} = \arg\min_\beta \|W_S(X\beta - Y)\|_2^2$ which is a weighted linear regression problem on $|S|$ examples. The works of [9, 13] show that resulting linear function produces a $(1 + \epsilon)$ multiplicative approximation to the optimal squared-loss:

$$\|X\widehat{\beta} - Y\|_2^2 \leq (1 + \epsilon)\inf_{\beta \in \mathbb{R}^d}\|X\beta - Y\|_2^2 \tag{8}$$

with constant probability. The weight matrix $W_S$ returned by leverage score sampling is constructed as follows: Letting $U\Sigma V^T$ denote the singular value decomposition of $X$, the algorithm iterates over all the points in $X$ and includes a point $x$ in $S$ with probability $p(x) \triangleq \min\left\{1, \frac{m}{d}\|U(x)\|_2^2\right\}$ proportional to its leverage score $\|U(x)\|_2^2$, where $U(x)$ is the row in $U$ corresponding to the point $x \in X$. The corresponding diagonal entry of $W_S$ is set as $\frac{1}{\sqrt{p(x)}}$. Since the points in $X_{\text{lab}}$ are labeled, the number of times the algorithm queries the oracle to label a point is equal to $|S \cap X_{\text{un}}|$. By definition of the sampling probabilities, this is proportional to the sum of the leverage scores across the points in $X_{\text{un}}$:

$$\mathbb{E}\left[|S \cap X_{\text{un}}|\right] = \frac{m}{d}\sum_{x \in X_{\text{un}}}\|U(x)\|_2^2 = \frac{\log(d)}{\epsilon}\sum_{x \in X_{\text{un}}}\|U(x)\|_2^2 \tag{9}$$

In Lemma 5, we prove that the term $\sum_{x \in X_{\text{un}}}\|U(x)\|_2^2$ equals the reduced rank, $\mathbf{R}_X$ defined as $\text{Tr}\left(\left(X_{\text{lab}}^T X_{\text{lab}} + X_{\text{un}}^T X_{\text{un}}\right)^{-1} X_{\text{un}}^T X_{\text{un}}\right)$. Combining with eq. (9) results in the equation,

$$\mathbb{E}\left[|S \cap X_{\text{un}}|\right] = \frac{\mathbf{R}_X \log(d)}{\epsilon} \tag{10}$$

In particular, by Markov's inequality, with large enough constant probability, leverage score sampling makes at most $O\left(\frac{\mathbf{R}_X \log(d)}{\epsilon}\right)$ label queries (of points in $X_1$). Combining with the guarantee on the quality of the solution returned by leverage score sampling in eq. (8) results in the following guarantee.

**Theorem 1.** *For $0 < \epsilon < 1$, the $\epsilon$-query complexity of leverage score sampling for SSAR is upper bounded by $O(\frac{\mathbf{R}_X \cdot \log(d)}{\epsilon})$.*

In order to further motivate $\mathbf{R}_X$ we next turn to the ridge regression problem: Here, it turns out that $\mathbf{R}_X$ equals the *statistical dimension* of $X_{\text{un}}$, defined as:

$$\text{sd}_\lambda(X_{\text{un}}) \triangleq \sum_{i=1}^{\text{rank}(X_{\text{un}})} \frac{1}{1 + \lambda/\sigma_i^2(X_{\text{un}})}, \tag{11}$$

where $\sigma_i(X_{\mathrm{un}})$ is the $i^{th}$ largest singular value of $X_{\mathrm{un}}$. This can be seen by plugging in $X_{\mathrm{lab}}$ as $\sqrt{\lambda}I$ in eq. (3) and expanding $X_{\mathrm{un}}$ using its singular value decomposition as $U_1 \Sigma_1 V_1$.

In comparison, in kernel ridge regression, $\mathrm{R}_X$ evaluates to the *effective dimension*, defined as:

$$d_\lambda(K) \triangleq \sum_{i=1}^{\mathrm{rank}(K)} \frac{1}{1 + \lambda/\sigma_i(K)}, \tag{12}$$

where the $\sigma_i$'s are the eigenvalues of $K$. This yet again follows by plugging the choice of $X_{\mathrm{un}}$ and $X_{\mathrm{lab}}$ in the kernel ridge regression setting from eq. (7).

The common theme in the prior sampling based approaches [1] for kernel ridge regression, as well as the bound on leverage score sampling established for SSAR (Theorem 1) is that they face a logarithmic dependence in the dimension parameter $d$ or $N$ (in the case of kernel ridge regression) which are often large in practice. Indeed, we show that this logarithmic dependence is unavoidable in the worst case for leverage score sampling.

**Theorem 2.** *There exists an instance of SSAR such that the $\epsilon$-query complexity of leverage score sampling for any constant $\epsilon$ is $\geq \frac{1}{4}\mathrm{R}_X \log(d)$.*

In this paper, we subsume previous results and show an algorithm with an $\epsilon$-query complexity of $O\left(\frac{\mathrm{R}_X}{\epsilon}\right)$ for SSAR. This shows that it is possible to derive statistical guarantees for active ridge / kernel ridge regression that respectively depend *only* on the statistical dimension / effective dimension.

## 4 Algorithm

In this section, we design a polynomial time algorithm (Algorithm 1) for SSAR and prove that the $\epsilon$-query complexity of the algorithm is $O(\frac{\mathrm{R}_X}{\epsilon})$.

The core construction of the algorithm is based on an iterative mechanism known as an $\epsilon$-*well balanced sampling procedure*, which decides which points to query the labels of points. At an intuitive level, an $\epsilon$-well balanced sampling procedure guarantees that the points sampled so far are well representative of the overall dataset (in a weighted sense), while at the same time guaranteeing that a certain weighted condition number, related to the error incurred by least squares regression is small.

Our main algorithm, Algorithm 1 follows the template of [6] which $(i)$ repeatedly queries the labels of points in an adaptive manner using an $\epsilon$-well balanced sampling procedure $\mathsf{Alg}$, and $(ii)$ solves a weighted linear regression problem on the set of points sampled in step $(i)$.

---

**Algorithm 1:** ASURA (Active semi-SUpervised Regression Algorithm)

---

**Input:** $X \in \mathbb{R}^{(n_{\mathrm{un}}+n_{\mathrm{lab}}) \times d}$; accuracy parameter $\epsilon$; any well-balanced sampling procedure $\mathsf{Alg}$ (cf. Definition 1 or Algorithm 2);

1 Run $\mathsf{Alg}(X, \epsilon)$ to return a subset of points $\{x_1, \cdots, x_m\}$ and weights $\{w_1, \cdots, w_m\}$.;
   // $m$ is determined internally by $\mathsf{Alg}$
2 Query the labels $y_i$ of $x_1, \cdots, x_m$ for $x_i \in X_{\mathrm{un}}$. ;
   // the label query cost of the algorithm

**Output:** $\widehat{\beta} \leftarrow \arg\min_{\beta \in \mathbb{R}^d} \sum_{i=1}^m w_i(\beta^T x_i - y_i)^2$.

---

We first begin by defining the notion of a well-balanced sampling procedure.

**Definition 1** (well-balanced sampling procedure). *[6, Def 2.1] A well-balanced sampling procedure is a randomized algorithm $\mathsf{Alg}(X, \epsilon)$ that outputs a set of points $\{x_1, \cdots, x_m\}$ and weights $\{w_1, \cdots, w_m\}$ as follows: in each iteration $i = 1, \cdots, m$, the algorithm chooses a distribution $D_i$ over points in $X$ to sample $x_i \sim D_i$. Let $D$ be the uniform distribution over all points in $X$. Then, $\mathsf{Alg}$ is said to be an $\epsilon$-**well-balanced sampling procedure** if it satisfies the following two properties with probability $\geq 3/4$,*

*1. The matrix $Z = \mathsf{diag}(\{\sqrt{w_i}\}_{i=1}^m) U_{[x_i:x_m]} \in \mathbb{R}^{m \times d}$, where $U\Sigma V^T$ is the SVD of $X$ and $U_{[x_i:x_m]}$ are the rows of $U$ corresponding to the points $\{x_1, \cdots, x_m\}$, satisfies:*

$$\frac{3}{4}I \preceq Z^T Z \preceq \frac{5}{4}I \tag{13}$$

*Equivalently, For every $\beta \in \mathbb{R}^d$, $\sum_{i=1}^m w_i \langle \beta, x_i \rangle^2 \in \left[ \frac{3}{4}, \frac{5}{4} \right] \mathbb{E}_{x \sim D} \left[ \langle \beta, x \rangle^2 \right]$.*

2. *For each $i \in [m]$, define $\alpha_i = \frac{D_i(x_i)}{D(x_i)} w_i$. Then, Alg must satisfy $\sum_{i=1}^m \alpha_i = O(1)$ and $\alpha_i K_{D_i} = O(\epsilon)$, where $K_{D_i}$ is the re-weighted condition number:*

$$K_{D_i} = \sup_x \left\{ \sup_{\beta \in \mathbb{R}^d} \left\{ \frac{D_i(x)}{D(x)} \cdot \frac{\langle \beta, x \rangle^2}{\mathbb{E}_{x' \sim D} \left[ \langle \beta, x' \rangle^2 \right]} \right\} \right\} \tag{14}$$

In [6] it is shown that any sampling procedure in step $(i)$ of Algorithm 1 which is *well-balanced* (Definition 1) guarantees that the resulting solution is approximately optimal.

**Theorem 3.** *[6, Theorem 2.3] For any $0 < \epsilon < 1$, instantiate Algorithm 1 with any $\epsilon$-well balanced sampling procedure $\mathsf{Alg}(X, \epsilon)$. Denoting $\widehat{\beta}$ as the output of Algorithm 1, with constant probability,*

$$\|X\widehat{\beta} - Y\|_2^2 \le (1 + O(\epsilon)) \min_{\beta \in \mathbb{R}^d} \|X\beta - Y\|_2^2 \tag{15}$$

Theorem 3 shows that to solve the SSAR problem, it suffices to design a well-balanced sampling procedure which queries a minimal number of labels of datapoints in the unlabeled set, $X_{\text{un}}$. We now present Algorithm 2 and show that it indeed satisfies both of these criteria. The algorithm is parameterized by $\gamma$, chosen to be $\triangleq \sqrt{\epsilon}/C_0$ for a sufficiently large constant $C_0 > 0$.

**Theorem 4.** *Algorithm 2 is an $\epsilon$-well-balanced sampling procedure, where $0 < \epsilon < 1$. Furthermore, Algorithm 2 samples the labels of at most $O\left( \frac{R_X}{\epsilon} \right)$ points in $X_{un}$.*

---

**Algorithm 2:** $\epsilon$-well balanced sampling procedure for SSAR

**Input:** $X \in \mathbb{R}^{n \times d}$.;
1 **Initialization:** $j = 0$; $\gamma = \sqrt{\epsilon}/C_0$; $u_0 = 2d/\gamma$; $l_0 = -2d/\gamma$; $A_0 = 0$.;
2 **while** $(u_j - l_j) + \sum_{i=0}^{j-1} \Phi_i^{\text{Id}} < 8d/\gamma$ **do**
3      Define $\Phi_j^{\text{Id}} = \mathsf{Tr}\left( (u_j I - A_j)^{-1} + (A_j - l_j I)^{-1} \right)$.;
4      Set coefficient $\alpha'_j = \gamma/\Phi_j^{\text{Id}}$;
5      Sample a point from the multinomial distribution on $X$ which assigns probability to $x$ as,

$$p_x \triangleq \frac{U(x)^T \left( (u_j I - A_j)^{-1} + (A_j - l_j I)^{-1} \right) U(x)}{\Phi_j^{\text{Id}}} \tag{16}$$

     `// Define the sampled point x as x_j and p_x as p_j`
6      Update $A_{j+1} \leftarrow A_j + \frac{\gamma}{\Phi_j^{\text{Id}}} \frac{1}{p_j} U(x_j)(U(x_j))^T$;
7      Define the weight $w'_j \leftarrow \frac{\gamma}{\Phi_j^{\text{Id}}} \frac{1}{p_j}$.;
8      Update $u_{j+1} \leftarrow u_j + \frac{\gamma}{1 - 2\gamma} \frac{1}{\Phi_j^{\text{Id}}}$ and $l_{j+1} \leftarrow l_j + \frac{\gamma}{1 + 2\gamma} \frac{1}{\Phi_j^{\text{Id}}}$.;
9      $j \leftarrow j + 1$.;
10 **end**
11 Assign $m = j$.;
12 Define $\text{mid} \triangleq \frac{u_m + l_m}{2}$ and for each $j$, set the coefficient $\alpha_j = \frac{\alpha'_j}{\text{mid}}$ and the weight $w_j = \frac{w'_j}{\text{mid}}$.;
**Output:** $\{x_1, \cdots, x_m\}$; $\{w_1, \cdots, w_m\}$.;

---

Algorithm 2 is a variant of the Randomized BSS algorithm of [16] for spectral sparsification (eq. (13)) with an updated stopping criterion, which we discuss in Remark 1. In each iteration, the algorithm maintains a matrix $A$ which up to a scaling factor is the same as the matrix $Z^T Z$ in Definition 1, and an upper and lower threshold $u$ and $l$ which are updated in an iterative manner. The update rule is guaranteed to maintain the invariant $lI \preceq A \preceq uI$ in every iteration, which is ensured by the adaptive sampling distribution in eq. (16). Over iterations, the algorithm shows that the upper and lower thresholds $u$ and $l$ can be brought closer to each other which sandwich $Z^T Z$ (equal to $A$ up to scaling) until eq. (13) is finally satisfied. It turns out that the desirable quality for is that $u$ and $l$ progressively become larger (which dictates the scaling factor relating $A$ and $Z^T Z$), while at the same time gap between the two, $u - l$, remains small. The proof of correctness of Algorithm 2 requires two ingredients:

1. Algorithm 2 is indeed an $\epsilon$-well balanced sampling procedure. (Section 4.1), and

2. The number of label queries made by Algorithm 2 is bounded by $O\left(\mathrm{R}_X/\epsilon\right)$ (Section 4.2).

**Remark 1.** *Notice the important difference in the stopping criterion of Algorithm 2 compared to that of the Randomized BSS Algorithm of [16] which is just the condition $u_j - l_j < 8d/\gamma$. The more aggressive stopping criterion can be used to show that Algorithm 2 terminates within $2d/\gamma^2$ iterations of the while loop in Step 2 almost surely, while the Randomized BSS algorithm [16] terminates within $2d/\gamma^2$ iterations only with constant probability. This behavior is critical in bounding the number of label queries made by Algorithm 2.*

## 4.1  Algorithm 2 is an $\epsilon$-well-balanced sampling procedure

We first address the question of showing that Algorithm 2 is an $\epsilon$-well balanced sampling procedure. We provide a brief sketch which addresses the first condition, eq. (13). First observe that the stopping criterion of the algorithm is the first time $j$ when $\sum_{i=0}^{j-1} \frac{4\gamma^2}{\Phi_i^{\mathrm{Id}}(1-4\gamma^2)} + \sum_{i=0}^{j-1} \Phi_i^{\mathrm{Id}} > \frac{8d}{\gamma}$. Note that each term in the summation, $\frac{4\gamma^2}{\Phi_i^{\mathrm{Id}}(1-4\gamma^2)} + \Phi_i^{\mathrm{Id}}$ is at least $2\sqrt{\frac{4\gamma^2}{\Phi_i^{\mathrm{Id}}(1-4\gamma^2)} \times \Phi_i^{\mathrm{Id}}} \geq 4\gamma$ by the AM-GM inequality. Therefore, if the number of terms in the summation is at least $\frac{8d}{\gamma} \times \frac{1}{4\gamma} = \frac{2d}{\gamma^2}$ the stopping criterion must certainly be violated. This results in a bound on the number of iterations before the while loop is terminated in Algorithm 2, defined as $m$.

**Lemma 1.** *Almost surely, $m \leq 2d/\gamma^2$.*

While $m$ is upper bounded by $2d/\gamma^2$, the key result of this section is that the number of iterations of by the algorithm *also satisfies $m \geq cd/\gamma^2$* with constant probability, for a sufficiently small constant $c$. The implication of the algorithm not terminating too soon is that in the upper and lower boundaries maintained in the algorithm are updated over a large number of iterations and therefore the eigenvalues of the matrix $A$ are very tightly sandwiched, proving the eigenvalue condition in eq. (13). We flesh out the details below.

Note that in each iteration, the upper boundary $u_j$ increments by $\frac{\gamma}{1-2\gamma} \frac{1}{\Phi_j^{\mathrm{Id}}}$. As we show in the Appendix A.3 (Lemma 7), $\Phi_j^{\mathrm{Id}}$ is $\geq \gamma/2$ almost surely and therefore $u_j$ increments by at least a constant $\frac{\gamma}{1-2\gamma} \times \frac{2}{\gamma} \geq 2$ in each iteration of Algorithm 2. Since the algorithm does not terminate too soon and $m \geq cd/\gamma^2$ with constant probability, in the final iteration, $u_m$ is also $\geq cd/\gamma^2 \times 2$ with constant probability. More precisely, we have the following result.

**Lemma 2.** *For $\gamma < 1/4$ and any $0 \leq p < 1$, with probability at least $1 - p$, $u_m \geq p^2 d/8\gamma^2$.*

We next relate the matrix $A$ maintained in the algorithm and the matrix $Z^T Z$ in Definition 1. When Algorithm 2 terminates, the matrix $A_m$ stored is equal to $\sum_{j=0}^{m-1} \frac{\gamma}{\Phi_j^{\mathrm{Id}}} \frac{1}{p_j} U(x_j)(U(x_j))^T$. From the definition of mid and $w_j$ in Algorithm 2, it is a quick calculation to show that $Z^T Z = \frac{1}{\mathrm{mid}} A = \frac{2}{u_m + l_m} A_m$. Furthermore, since the invariant of Algorithm 2 guarantees that $l_m I \preceq A_m \preceq u_m I$, we have that $\frac{2l_m}{u_m + l_m} \preceq Z^T Z \preceq \frac{2u_m}{u_m + l_m}$. Thus to prove eq. (13) it suffices to show that $\frac{u_m - l_m}{u_m + l_m}$ is small.

By the stopping criterion of the algorithm, we are guaranteed that the gap, $u_m - l_m \approx u_{m-1} - l_{m-1} \leq 8d/\gamma$; formally we show that $u_m - l_m \leq 9d/\gamma$ in Appendix A.4 (Lemma 9), implying that $u_m$ and $l_m$ are close to each other. Moreover from Lemma 2, with constant probability, $l_m \approx u_m = \Omega(d/\gamma^2)$. Thus, the ratio $\frac{u_m - l_m}{u_m + l_m} = O(\gamma)$ and satisfies eq. (13) for sufficiently small $\gamma$.

**Lemma 3.** *With probability at least $3/4$, $(1 - O(\gamma))I \preceq A \preceq (1 + O(\gamma))I$.*

We defer the proof of the fact that Algorithm 2 satisfies the second condition in Definition 1 to Appendix A (Lemma 12) since it largely follows from the analysis in [6].

## 4.2  Bounding the number of label queries made by Algorithm 2

Finally, we upper bound the number of times Algorithm 1 queries the labels of points in the unlabeled set, $X_{\mathrm{un}}$. We denote this quantity by $\mathfrak{lq}(X_{\mathrm{un}})$.

First observe that the number of label queries, $\mathfrak{lq}(X_{\mathrm{un}})$, is at most the number of iterations in which the well-balanced sampling procedure $\mathsf{Alg}$ in Line 1 of Algorithm 1 samples a point in $X_{\mathrm{un}}$. The

former is smaller if a point is sampled more than once by Alg . Indeed,

$$\mathbb{E}\left[\mathsf{lq}(X_{\text{un}})\right] \leq \mathbb{E}\left[\sum_{j=0}^{m-1}\sum_{x \in X_{\text{un}}} \mathbb{I}_x^{(j)}\right], \tag{17}$$

where $m$ is the total number of iterations the algorithm runs for, and $\mathbb{I}_x^{(j)}$ is an indicator capturing which is 1 if $x$ is sampled in Step 5 of Algorithm 2 in iteration $j$ and 0 otherwise. Note that $\mathbb{E}[\mathbb{I}_x^{(j)}] = \mathbb{E}[p_x^{(j)}]$ is the probability of sampling the point $x$ as defined in eq. (16). Recall that in Lemma 1 we show that $m$ is upper bounded by $\frac{2d}{\gamma^2}$ almost surely. By linearity of expectation,

$$\mathbb{E}[\mathsf{lq}(X_{\text{un}})] \leq \sum_{j=0}^{2d/\gamma^2 - 1}\sum_{x \in X_{\text{un}}} \mathbb{E}\left[p_x^{(j)}\right]. \tag{18}$$

**Remark 2.** *Note that the bound on the expected number of label queries in eq.* (17) *is the sum of non-negative random variables ($\mathbb{I}_x^{(j)}$) which are correlated with the number of terms in the summation ($m$). The almost sure upper bound on $m$ enables this correlation to be decoupled, which is not easily possible with a constant or even high probability upper bound on $m$, as guaranteed by the Randomized BSS algorithm [16] (discussed in Remark 1).*

Before studying $\sum_{j=0}^{2d/\gamma^2-1}\sum_{x \in X_{\text{un}}} \mathbb{E}\left[p_x^{(j)}\right]$, we first introduce some relevant notation.

**Definition 2.** *For a PSD matrix $M$, define the potential function $\Phi_j^M = \mathsf{Tr}(M(u_j I - A_j)^{-1} + M(A_j - l_j I)^{-1}))$. $u_j$, $l_j$ and $A_j$ are as defined in Algorithm 2.*

Simplifying from definition of $p_x^{(j)}$ results in the following lemma.

**Lemma 4.** *In iteration $j = 0, \cdots, m-1$, $\sum_{x \in X_{un}} p_x^{(j)} = \frac{\Phi_j^D}{\Phi_j^{Id}}$, where $D = \sum_{x \in X_{un}} U(x)(U(x))^T$.*

As we show in Appendix A (Lemma 7), the term in the denominator, $\Phi_j^{\text{Id}}$, is at least $\Phi_j^{\text{Id}} \geq \gamma/2$ almost surely. By plugging in Lemma 4 into eq. (18), the expected number of labels of points $X_{\text{un}}$ queried by Algorithm 2 is upper bounded by,

$$\mathbb{E}[\mathsf{lq}(X_{\text{un}})] \leq \frac{2}{\gamma}\sum_{j=0}^{2d/\gamma^2-1} \mathbb{E}[\Phi_j^D] \tag{19}$$

Over the course of the time, one expects the gap between $u_j$ and $l_j$ to grow apart (from the update rule on line 8 of Algorithm 2), while $A_j$ remains tightly sandwiched in between the two bounds. It turns out that a consequence of this fact, the expectation of the potential, $\mathbb{E}\left[\Phi_j^D\right]$ is a decreasing function in $j$. We formally prove this result in Appendix A.3 (Lemma 8). From eq. (19), this results in the following upper bound on the number of label queries,

$$\mathbb{E}[\mathsf{lq}(X_{\text{un}})] \leq \frac{4d}{\gamma^3}\mathbb{E}\left[\Phi_0^D\right] = \frac{4\mathsf{Tr}(D)}{\gamma^2} \tag{20}$$

where the last equation, $\mathbb{E}[\Phi_0^D] = \mathbb{E}[\mathsf{Tr}(D(u_0 I - A_0)^{-1} + D(A_0 - l_0 I)^{-1})] = \frac{\gamma \mathsf{Tr}(D)}{d}$ is a short calculation from its definition using the fact that $u_0 = -l_0 = \frac{2d}{\gamma}$ and $A_0 = 0$. The final lemma relates $\mathsf{Tr}(D)$ to the reduced rank $\mathsf{R}_X$.

**Lemma 5** (Relating $D$ to $\mathsf{R}_X$). *With $D = \sum_{x \in X_{un}} U(x)(U(x))^T$, $\mathsf{Tr}(D) = \mathsf{R}_X$.*

Putting together Lemma 5 with eq. (20), and noting the definition $\gamma = \sqrt{\epsilon}/C_0$ results in the final bound on the number of label queries made by Algorithm 2.

$$\mathbb{E}[\mathsf{lq}(X_{\text{un}})] \lesssim \frac{4\mathsf{R}_X}{\epsilon}. \tag{21}$$

**Remark 3.** *In the case of ridge regression, Algorithm 1 can be used to construct a weak-coreset for $\beta^*$. The advantage of the algorithm over [13] is that the weak coreset can be constructed without having oracle access to the set of all labels. In particular, our algorithm returns a coreset which uses $O(d\mathsf{sd}_\lambda/\epsilon)$ bits of memory: up to log-factors, the weight matrix $W_S$ requires $\frac{d}{\epsilon}$ bits of memory to store and the $\mathsf{sd}_\lambda/\epsilon$ points queried in $X_{un}$ require $\frac{d\mathsf{sd}_\lambda}{\epsilon}$ bits of memory to store. Solving: $\arg\min_{\beta \in \mathbb{R}^d} \|W_S X \beta - W_S Y\|_2^2$ returns a $(1 + \epsilon)$ approximate optimizer of the loss. The memory requirement of our algorithm matches that proposed in [13], while only querying a few labels.*

# 5 Instance-dependent lower bound for distributional SSAR

In this section we introduce a distributional or *inductive* [6] formulation of semi-supervised active regression as a special case of the agnostic formulation discussed earlier.

**Definition 3** (Distributional SSAR)**.** *The setup here is identical to the agnostic setting, but with the labels revealed to the learner corrupted by independent stochastic noise. In particular, at training time, each time the label of $x \in X$ is observed (either in the labeled dataset, or by explicitly querying) a noisy version of the ground-truth label, $f(x)$, is revealed. Namely, $y = f(x) + Z$ is observed, where the training time noise $Z \sim \mathcal{N}(0, \sigma_x^2)$ [2]. The objective is to minimize the generalization error,*

$$\mathbb{E}\left[\frac{1}{|X|}\sum\nolimits_{x \in X}\left(\langle\beta, x\rangle - f(x) - Z'_x\right)^2\right] \tag{22}$$

*where for each $x \in X$, $Z'_x \sim \mathcal{N}(0, \sigma_x^2)$ is sampled independent of the noise at training time.*

Note that the distributional SSAR problem is a special case of the agnostic formulation. This just corresponds to having infinitely many copies of the labeled and unlabeled datasets, where the label in each copy is the ground truth corrupted by a different realization of the stochastic noise. Therefore, it may be easily verified that the guarantee of ASURA (Theorem 3) carries over here.

**Corollary 1.** *The ASURA algorithm for the distributional SSAR can be implemented efficiently (polynomial in $d$, the support size of $X$) for the distributional SSAR problem, requiring to compute SVDs over a single copy of the matrix $X$. Furthermore, the guarantee of ASURA still carries over: the algorithm queries the labels of at most $O(\mathrm{R}_X/\epsilon)$ points, and guarantees to achieve a $(1 + \epsilon)$ approximation to the best achievable generalization loss (eq. (22)).*

It turns out that in the distributional setting, one may also come up with an instance-dependent sample complexity lower bound, where the input design $X$ is fixed, but the adversary has the power to control the ground truth labels $f(\cdot)$ as well as the noise variances $\sigma_x^2$ in designing the instance.

**Theorem 5.** *For any $0 < \epsilon < 1$, $d$ and any $\lambda > 0$. Suppose $X$ satisfies the condition, $\sigma_{\min}\left(X(X^TX)^{-1}X_{lab}^T\right) \geq \frac{\epsilon}{1-\epsilon}$. In the inductive setting, for each $X$ and learner there exists an instance of SSAR where if $\widehat{\beta}$ returned by the learner satisfies $\mathbb{E}\left[\|X\widehat{\beta} - Y\|_2^2\right] \leq (1 + \epsilon)\min_{\beta \in \mathbb{R}^d}\mathbb{E}\left[\|X\beta - Y\|_2^2\right]$ must query the labels of at least $\Omega(\frac{\mathrm{R}_X}{\epsilon})$ points.*

The proof of this lower bound is quite complicated because of both the adaptive nature of the problem, as well the requirement for an instance-dependent bound. At a high level, we use an hypothesis testing based construction (Assouad's lemma) with a careful choice of hypothesis space. The main challenge is choosing the underlying instance in a way to recover the reduced rank as the measure of complexity, which has a very delicate dependence on (the spectrum of) $X_{\text{lab}}$ and $X_{\text{un}}$. In the interest of space, we defer the details of th proof to Appendix A.5. This result also shows that there is no need to have any sort of commutation structure between the labelled and unlabelled data matrices $X_{\text{lab}}$ and $X_{\text{un}}$ for the reduced rank to remain as the right instance dependent lower bound.

# 6 Conclusion

We introduce the semi-supervised active learning problem and prove a query complexity upper bound of $\mathrm{R}_X/\epsilon$ for linear regression. In the special case of active ridge regression, this implies a sample complexity upper bound of $O(\mathrm{sd}_\lambda/\epsilon)$ labels which we prove is optimal. The problem also generalizes active kernel ridge regression, where we show a query complexity upper bound of $O(d_\lambda/\epsilon)$ improving the results of [1]. It is an open question to extend these results to other loss functions and to design near-linear time algorithms for the problem.

# 7 Acknolwedgements

NR was partially supported by NSF Grants IIS-1901252, and CCF-1909499.

---

[2]one may instead assume that the noise is subgaussian with variance $\sigma_x^2$. For ease of exposition we stick to Gaussian noise.

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
