# A  Proofs of main results

## A.1  Proof of Theorem 2

Consider an instance of semi-supervised active regression with the labeled dataset $X_{\text{lab}}$ being empty, and with the unlabeled dataset composed of $k$ copies of the standard basis vector $e_i$ for each $i = 1, \cdots, d$. $k$ will be later be taken to $\infty$ and can be thought of as being very large. The labels are assumed to be generated by a random linear function. Namely, we consider a family of instances where for each $i$, $\beta_i^* \sim \mathcal{N}(0, 1)$ and the ground truth labels of the points aligned with $e_i$ are set as $\beta_i^*$. All instances in this family therefore satisfy realizability, namely, $\|X\beta^* - Y\|_2^2 = 0$. Moreover, on this family of instances, the reduced rank $\mathsf{R}_X = \mathsf{Tr}\left( \left( X_{\text{lab}}^T X_{\text{lab}} + X_{\text{un}}^T X_{\text{un}} \right)^{-1} X_{\text{un}}^T X_{\text{un}} \right) = d$, since there are no prior labeled datapoints.

The total number of points sampled by leverage score sampling, $m$, is taken as a parameter. We show that if $m \leq cd\log(d)$ for a sufficiently small constant $C$, with high probability, the linear function returned by the algorithm is poor in the sense that $\|X\widehat{\beta} - Y\|_2^2 > 0$ (and thus the approximation ratio, $(1 + \epsilon)$, is no longer bounded for the algorithm).

First, we compute the sampling probabilities for each point $x \in X$. Recall that Leverage score sampling assigns the probability of sampling the point $x$ as $\min\{1, \frac{m}{d}\|U(x)\|_2^2\}$. In the given instance, for each $x \in X$, $\|U(x)\|_2^2 = \frac{1}{k}$. All in all, each point $x$ is sampled with probability $\frac{m}{dk}$. As $k \to \infty$, the number of labels of points sampled along each direction $e_i$ tends in distribution to a Poisson distributed random variable, in particular, distributed $\sim \mathsf{Poi}(m/d)$. Finally, the number of times points aligned with $e_i$ are sampled are independent across $i$. Therefore, the probability that there exists at least one $i$ such that leverage score sampling samples no points along that direction $e_i$ is, $1 - \left(1 - e^{-m/d}\right)^d$. If $m \leq \frac{1}{2}d\log(d)$, then,

$$1 - \left(1 - e^{-m/d}\right)^d = 1 - \left(1 - \frac{1}{\sqrt{d}}\right)^d \geq 1 - e^{-\sqrt{d}} \tag{23}$$

Therefore, with high probability no points are sampled along at least one of the directions $e_i$. As a consequence, if $m \leq \frac{1}{2}d\log(d)$, leverage score sampling algorithm never observes $\beta_i^*$ for at least one index $i \in [d]$ (say, $i_0$) with probability $\geq 1 - e^{-\sqrt{d}}$. The prediction of the algorithm on points aligned with $e_{i_0}$ is statistically independent of the true label $\beta_{i_0}^* \sim \mathcal{N}(0, 1)$ - in other words, the learner has no way of guessing $\beta_{i_0}^*$. Therefore,

$$\|X\beta - Y\|_2^2 \geq k\mathbb{E}\left[(\widehat{\beta}_i - \beta_{i_0}^*)^2\right] \geq k\left(\mathbb{E}\left[(\beta_{i_0}^*)^2\right]\right) = k \tag{24}$$

where the expectation is taken only over $\beta_{i_0}^*$. Since for this family of instances $\mathsf{R}_X = d$, this shows the existence of a semi-supervised active regression instance such that if leverage score sampling is run with $m \leq \frac{1}{2}\mathsf{R}_X\log(d)$, with probability $\geq 1 - e^{-\sqrt{d}}$, the approximation factor admit by leverage score sampling is unbounded. More importantly, since there are no prior labeled points in the dataset, the number of labels queried by the algorithm, in expectation, is $d \times \frac{m}{d} = m = \frac{1}{2}\mathsf{R}_X\log(d)$. Moreover, by concentration of the sum of independent random variables, the number of labels queried is no smaller than $\frac{1}{4}\mathsf{R}_X\log(d)$ with probability $\geq 1 - e^{-\Omega(d)}$. Therefore, by union bounding, with probability $\geq 1 - e^{-\sqrt{d}} - e^{-d}$, both (i) the approximation factor admit by leverage score sampling is $\infty$, and (ii) the number of labels sampled by leverage score sampling is $\geq \frac{1}{4}\mathsf{R}_X\log(d)$.

## A.2  Proof of Theorem 3

First we restate and prove Lemma 5 which relates the trace of the matrix $D$ to the $\mathsf{R}_X$ parameter.

**Lemma 5** (Relating $D$ to $\mathsf{R}_X$). *With $D = \sum_{x \in X_{un}} U(x)(U(x))^T$, $\mathsf{Tr}(D) = \mathsf{R}_X$.*

*Proof.* Observe that $X = \begin{bmatrix} X_{\text{un}} \\ X_{\text{lab}} \end{bmatrix} = U\Sigma V^T$. Therefore, $D = U^T S U$ where $S$ is a diagonal matrix with 1's on rows corresponding to $x \in X_{\text{un}}$ and 0's otherwise. Observe that $X^T S X =$

$V\Sigma U^T S U\Sigma V^T$. Moreover, observe that $X^T S X = \begin{bmatrix} X_{\text{un}}^T & X_{\text{lab}}^T \end{bmatrix} S \begin{bmatrix} X_{\text{un}} \\ X_{\text{lab}} \end{bmatrix} = X_{\text{un}}^T X_{\text{un}}$. Therefore,

$$D = U^T S U = (V\Sigma)^{-1} X_{\text{un}}^T X_{\text{un}} (\Sigma V^T)^{-1} \tag{25}$$

Tracing both sides and using the commutativity of the trace operator,

$$\mathsf{Tr}(D) = \mathsf{Tr}\left( \left( \left(V^T\right)^{-1} \Sigma^{-2} V^{-1} \right) X_{\text{un}}^T X_{\text{un}} \right) \tag{26}$$

$$= \mathsf{Tr}\left( \left(X^T X\right)^{-1} X_{\text{un}}^T X_{\text{un}} \right) \tag{27}$$

$\square$

## A.3 Upper bounding the number of points in $X_{\text{un}}$ sampled by Algorithm 2 (Theorem 4)

We first bound the number of points sampled in $X_{\text{un}}$ by Algorithm 2. We begin by re-stating Lemma 6 which explicitly computes the expected number of points sampled sampled by Algorithm 2 in terms of various potentials.

**Lemma 6.** *Recall that Algorithm 2 samples a subset of the $n_{un} + n_{lab}$ points in $X_{un} \cup X_{lab}$. The expected number of iterations the algorithm samples a point in $X_{un}$ is given by:* $\mathbb{E}[\mathsf{lq}(X_{un})] \leq \mathbb{E}\left[ \sum_{j=1}^{m-1} \frac{\Phi_j^D}{\Phi_j^{\text{Id}}} \right]$ *where $D = \sum_{x \in X_{un}} U(x)(U(x))^T$.*

*Proof.* From eq. (17), the number of unlabelled points sampled by the algorithm is upper bounded by

$$\mathbb{E}\left[\mathsf{lq}(X_{\text{un}})\right] \leq \mathbb{E}\left[ \sum_{j=0}^{m-1} \sum_{x \in X_{\text{un}}} p_x^{(j)} \right] \tag{28}$$

$$= \mathbb{E}\left[ \sum_{j=0}^{m-1} \sum_{x \in X_{\text{un}}} \frac{U(x)^T \left( (u_j I - A_j)^{-1} + (A_j - l_j I)^{-1} \right) U(x)}{\Phi_j^{\text{Id}}} \right] \tag{29}$$

$$= \mathbb{E}\left[ \sum_{j=0}^{m-1} \frac{\sum_{x \in X_{\text{un}}} \mathsf{Tr}\left( U(x) U(x)^T \left( (u_j I - A_j)^{-1} + (A_j - l_j I)^{-1} \right) \right)}{\Phi_j^{\text{Id}}} \right] \tag{30}$$

$$= \mathbb{E}\left[ \sum_{j=0}^{m-1} \frac{\mathsf{Tr}\left( \left( \sum_{x \in X_{\text{un}}} U(x) U(x)^T \right) \left( (u_j I - A_j)^{-1} + (A_j - l_j I)^{-1} \right) \right)}{\Phi_j^{\text{Id}}} \right] \tag{31}$$

$$= \mathbb{E}\left[ \sum_{j=0}^{m-1} \frac{\Phi_j^D}{\Phi_j^{\text{Id}}} \right] \tag{32}$$

where $D = \sum_{x \in X_{\text{un}}} U(x)(U(x))^T$ $\square$

Lemma 6 bounds the number of labels queried by Algorithm 2 among points in $X_{\text{un}}$. However the appearance of the potential $\Phi_j^{\text{Id}}$ in the denominator is challenging to bound, so we introduce another result to further upper bound this term.

**Lemma 7.** *In every iteration $0 \leq j < m$ of Algorithm 2, almost surely $\Phi_j^{\text{Id}} \geq \frac{1}{2}\gamma$.*

*Proof.* Note that for $j \in [m-1]$, $\Phi_j^{\text{Id}} = \mathsf{Tr}((u_j I - A_j)^{-1} + (A_j - l_j I)^{-1})$. Note that $A_j$ is a symmetric matrix. Suppose it is diagonalized as $U\Theta U^T$ where $\Theta = \mathsf{diag}(\theta_1, \cdots, \theta_d)$ are its eigenvalues. Then, $\Phi_j^{\text{Id}} = \sum_{t=1}^{d} \frac{1}{u_j - \theta_t} + \frac{1}{\theta_t - l_j}$. We show in Lemma 17 that $l_j I \preceq A_j \preceq u_j I$. With this constraint on the $\theta_t$'s, by minimizing, we obtain: $\frac{1}{u_j - \theta_t} + \frac{1}{\theta_t - l_j} \geq \frac{4}{u_j - l_j}$. Therefore, $\Phi_j^{\text{Id}} \geq \frac{4d}{u_j - l_j}$. Furthermore, by the stopping criterion of the algorithm, in every iteration $j < m$ of the algorithm, $u_j - l_j \leq \frac{8d}{\gamma}$. Therefore, for every $j = 0, 1, \cdots, m-1$, $\Phi_j^{\text{Id}} \geq \frac{1}{2}\gamma$. $\square$

Using Lemmas 6 and 7, we can bound the expected number of label queries made by Algorithm 2 as,

$$\mathbb{E}[\mathsf{lq}(X_{\text{un}})] \leq \frac{2}{\gamma} \mathbb{E}\left[\sum_{j=0}^{m-1} \Phi_j^D\right] \tag{33}$$

While this expression is nicer than that in lemma 6, it is still the case that the stopping time of the algorithm, $m$ and the potentials $\Phi_j^D$ are correlated in a complicated manner. Moreover, even if a high probability bound is known on $m$, it is not clear how to bound the expectation since the typical behavior of $\Phi_j^D$'s is not apparent. To decouple the $\Phi_j^D$'s with $m$, we instead show that $m$ is upper bounded almost surely. Since the $\Phi_j^D$'s are non-negative, this immediately results in an upper bound which is independent of $m$. Restating Lemma 1 below.

**Lemma 1.** *Almost surely, $m \leq 2d/\gamma^2$.*

*Proof.* Assuming that the algorithm has not terminated till the $(t+1)^{th}$ iteration, $u_t - l_t = u_0 - l_0 + \sum_{j=0}^{t-1} \frac{4\gamma^2}{\Phi_j^{\text{Id}}(1-4\gamma^2)} < \frac{8d}{\gamma}$ (this uses the fact that $u_{j+1} - u_j = \frac{\gamma}{\Phi_j^{\text{Id}}(1-2\gamma)}$ and $l_{j+1} - l_j = \frac{\gamma}{\Phi_j^{\text{Id}}(1+2\gamma)}$).
Observe that the event,

$$\{m \geq t\} = \{u_t - l_t < 8d/\gamma\} \stackrel{(i)}{=} \left\{\sum_{j=0}^{t-1} \frac{4\gamma^2}{\Phi_j(1-4\gamma^2)} + \sum_{j=0}^{t-1} \Phi_j^{\text{Id}} < \frac{4d}{\gamma}\right\} \stackrel{(ii)}{\subseteq} \left\{2\gamma \cdot t < \frac{4d}{\gamma}\right\} \tag{34}$$

where $(i)$ uses the fact that by definition, $u_{j+1} - l_{j+1} = u_j - l_j + \frac{\gamma}{\Phi_j(1-2\gamma)} - \frac{\gamma}{\Phi_j(1+2\gamma)}$ with $u_0 - l_0 = \frac{4d}{\gamma}$, and then cascading the sum to get an explicit form for $u_t - l_t$ $(ii)$ uses the AM-GM inequality. Therefore, with $t = \frac{2d}{\gamma^2}$, the event $\{m \geq t\}$ happens with probability 0. $\square$

Since $\Phi_j^D \geq 0$ almost surely, from eq. (33) and Lemma 1 the expected number of label queries made by Algorithm 2 is bounded by,

$$\mathbb{E}[\mathsf{lq}(X_{\text{un}})] \leq \frac{2}{\gamma} \sum_{j=0}^{2d/\gamma^2-1} \mathbb{E}\left[\Phi_j^D\right]. \tag{35}$$

To further simplify this expression, we bound $\mathbb{E}\left[\Phi_j^D\right]$. Indeed, in the following lemma we show that it is a decreasing function of $j$, so we have the inequality $\mathbb{E}\left[\Phi_j^D\right] \leq \mathbb{E}\left[\Phi_0^D\right]$.

**Lemma 8** (Bounding the potential). *For any fixed PSD matrix $M \succeq 0$, $\mathbb{E}[\Phi_{j+1}^M] \leq \mathbb{E}[\Phi_j^M]$.*

*Proof.* Recall that $\Phi_j^M = \text{Tr}(M(u_jI - A_j)^{-1} + \text{Tr}(M(A_j - l_jI)^{-1})$. $\Phi_{j+1}^M$ can be written as $\text{Tr}(M(u_{j+1}I - A_j - w_jw_j^T)^{-1}) + \text{Tr}(M(A_j + w_jw_j^T - l_{j+1}I)^{-1})$. Following a similar approach as BSS Lemma 3.3 and 3.4, invoking the Sherman-Morrison inversion formula,

$$\left(u_{j+1}I - A_j - w_jw_j^T\right)^{-1} = (u_{j+1}I - A_j)^{-1} + \frac{(u_{j+1}I - A_j)^{-1}w_jw_j^T(u_{j+1}I - A_j)^{-1}}{1 - w_j^T(u_{j+1}I - A_j)^{-1}w_j}. \tag{36}$$

Multiplying by $M$ and tracing both sides,

$$\text{Tr}\left(M\left(u_{j+1}I - A_j - w_jw_j^T\right)^{-1}\right) = \text{Tr}(M(u_{j+1}I-A_j)^{-1}) + \frac{\text{Tr}(M(u_{j+1}I - A_j)^{-1}w_jw_j^T(u_{j+1}I - A_j)^{-1})}{1 - w_j^T(u_{j+1}I - A_j)^{-1}w_j}. \tag{37}$$

Note that with probability 1, $w_jw_j^T \preceq \gamma(u_jI - A_j) \preceq \gamma(u_{j+1}I - A_j)$. Therefore, $w_j^T(u_{j+1} - A_j)^{-1}w_j \leq \gamma$. Therefore,

$$\text{Tr}\left(M\left(u_{j+1}I - A_j - w_jw_j^T\right)^{-1}\right) \leq \text{Tr}(M(u_{j+1}I-A_j)^{-1}) + \frac{\text{Tr}(M(u_{j+1}I - A_j)^{-1}w_jw_j^T(u_{j+1}I - A_j)^{-1})}{1 - \gamma}. \tag{38}$$

Finally, using linearity of expectation and noting that $\mathbb{E}\left[w_j w_j^T | A_j\right] = \frac{\gamma}{\Phi_j^{\mathrm{Id}}} I$, we have that,

$$\mathbb{E}\left[\mathsf{Tr}\left(M\left(u_{j+1}I - A_j - w_j w_j^T\right)^{-1}\right)\right] \leq \mathbb{E}\left[\mathsf{Tr}(M(u_{j+1}I - A_j)^{-1})\right] + \mathbb{E}\left[\frac{\gamma}{\Phi_j^{\mathrm{Id}}(1-\gamma)}\mathsf{Tr}(M(u_{j+1}I - A_j)^{-2})\right].$$
(39)

By a similar calculation as before,

$$\mathbb{E}\left[\mathsf{Tr}\left(M\left(A_j + w_j w_j^T - l_{j+1}I\right)^{-1}\right)\right] \leq \mathbb{E}\left[\mathsf{Tr}(M(A_j - l_{j+1}I)^{-1})\right] - \mathbb{E}\left[\frac{\gamma}{\Phi_j^{\mathrm{Id}}(1+2\gamma)}\mathsf{Tr}\left(M(A_j - l_{j+1}I)^{-2}\right)\right].$$
(40)

Note the difference from before, for $\gamma \leq \frac{1}{4}$, we use the inequality $w_j w_j^T \preceq 2\gamma(A_j - l_{j+1}I)$ which we derive in Lemma 16. This appears as the $1 + 2\gamma$ factor in the denominator of the second term in eq. (40).

Now observe that, $u_{j+1} - u_j = \frac{\gamma}{\Phi_j^{\mathrm{Id}}(1-2\gamma)} \geq \frac{\gamma}{\Phi_j^{\mathrm{Id}}(1-\gamma)}$ and $l_{j+1} - l_j = \frac{\gamma}{\Phi_j^{\mathrm{Id}}(1+2\gamma)}$. Therefore, adding eq. (39) and eq. (40) together,

$$\begin{aligned}
\mathbb{E}\left[\Phi_j^M\right] &\leq \mathbb{E}\left[\mathsf{Tr}(M(u_{j+1}I - A_j)^{-1} + M(A_j - l_{j+1}I)^{-1})\right] \\
&\quad + \mathbb{E}\left[(u_{j+1} - u_j)\mathsf{Tr}(M(u_{j+1}I - A_j)^{-2}) - (l_{j+1} - l_j)\mathsf{Tr}(M(A_j - l_{j+1}I)^{-2})\right]
\end{aligned}$$
(41)

Define $\Delta_u = u_{j+1} - u_j$ and $\Delta_l = l_{l+1} - l_j$ and for $t \in [0,1]$, define the function

$$f(t) = \mathsf{Tr}\left(M((u_j + \Delta_u t)I - A_j)^{-1} + M(A_j - (l_j + t\Delta_l)I)^{-1}\right).$$
(42)

Under the assumption $l_j I \preceq A_j \preceq u_j I$, the function $f(t)$ is convex in $t$. Therefore, $f(0) - f(1) \geq -\frac{\mathrm{d}f(t)}{\mathrm{d}t}\Big|_{t=1}$. In eq. (41) observe that the RHS is precisely $f(1) - \frac{\mathrm{d}f(t)}{\mathrm{d}t}\Big|_{t=1}$. Upper bounding this by $f(0)$, results in the equation

$$\mathbb{E}[\Phi_{j+1}^M] \leq \mathbb{E}\left[\mathsf{Tr}(M(u_j I - A_j)^{-1} + M(A_j - l_j I)^{-1})\right] = \mathbb{E}[\Phi_j^M].$$
(43)

$\square$

From eq. (35),

$$\mathbb{E}[\mathfrak{lq}(X_{\mathrm{un}})] \leq \frac{2}{\gamma}\sum_{j=0}^{2d/\gamma^2 - 1}\mathbb{E}\left[\Phi_j^D\right] \leq \frac{2}{\gamma}\frac{2d}{\gamma^2}\mathbb{E}\left[\Phi_0^D\right] = \frac{4d}{\gamma^3}\mathsf{Tr}(D(u_0 I - A_0)^{-1} + D(A_0 - l_0 I)^{-1}) \overset{(i)}{=} \frac{4d}{\gamma^3}\frac{\gamma R_X}{d} = \frac{4R_X}{\gamma^2}.$$
(44)

where $(i)$ uses the fact that $u_0 = \frac{2d}{\gamma}$ and $l_0 = -\frac{2d}{\gamma}$ and $\mathsf{Tr}(D) = R_X$ from Lemma 5.

This completes the bound on the number of labels queried by Algorithm 2. Next we move on to showing that Algorithm 2 is indeed an $\epsilon$-well-balanced sampling procedure which will complete the proof of Theorem 4.

### A.4 Algorithm 2 is $\epsilon$-well balanced sampling procedure

In order to satisfy the first property of Definition 1, we need to show that $Z^T Z$ is well conditioned and that its normalized eigenvalues lie in an interval $[1 - O(\gamma), 1 + O(\gamma)] \subseteq [3/4, 5/4]$ for sufficiently small $\gamma$. As we discuss in Section 4.1 of the paper, and revisit in more detail later, $Z^T Z = \frac{1}{(u_m + l_m)/2}A_m$, where $m$ is the number of iterations the while loop in Algorithm 2 runs for, and $A_j$ is as defined in Algorithm 2. Moreover, from Lemma 17, the eigenvalues of $A_j$ for any $j$ are bounded between $u_j$ and $l_j$ and when the algorithm terminates, the gap between $u_m$ and $l_m$ is $O(d/\gamma)$. Furthermore, from Lemma 2, with constant probability, $u_m$ is also lower bounded by $\Omega(d/\gamma^2)$. Thus, when $\gamma$ is not too large, $l_m \approx u_m = \Omega(d/\gamma^2)$ and $u_m - l_m = O(d/\gamma)$. These two conditions show that the eigenvalues of $Z^T Z = \frac{1}{(u_m + l_m)/2}A_m$ lie in the interval $[1 - O(\gamma), 1 + O(\gamma)]$ which is $\subseteq [3/4, 5/4]$ for any sufficiently small choice of $\gamma$ showing indeed that the first property for $\epsilon$-well balanced sampling procedures is satisfied by Algorithm 2. First we show the key result of this section that with constant probability $u_m$ is indeed lower bounded by $\Omega(d/\gamma^2)$.

**Lemma 2.** *For $\gamma < 1/4$ and any $0 \le p < 1$, with probability at least $1 - p$, $u_m \ge p^2 d/8\gamma^2$.*

*Proof.* First, observe that $u_m > \sum_{j=0}^{m-1} \frac{\gamma}{\Phi_j^{\mathrm{Id}}}$ and $\left(\sum_{j=0}^{t-1} \frac{1}{\Phi_j}\right)\left(\sum_{j=0}^{t-1} \Phi_j\right) \ge t^2$, we want to analyze

$$\Pr\left(\frac{p^2 d}{8\gamma^3} \le \frac{m^2}{\sum_{j=0}^{m-1} \Phi_j^{\mathrm{Id}}}\right) = \Pr\left(\sum_{j=0}^{m-1} \Phi_j^{\mathrm{Id}} \cdot \frac{p^2 d}{8\gamma^3} \le m^2\right) \tag{45}$$

$$\overset{(i)}{\ge} \Pr\left(\frac{p^2 d^2}{\gamma^4} \le m^2\right) \tag{46}$$

Where the last sufficient condition $(i)$ comes from the stopping criterion, which implies $\sum_{j=0}^{m-1} \Phi_j^{\mathrm{Id}} \le \frac{8d}{\gamma}$.

Now we prove an upper bound to $\Pr(m < g)$ where $g \triangleq \frac{pd}{\gamma^2}$.

$$\Pr\left(m < g\right) \overset{(i)}{=} \Pr\left(\sum_{j=0}^{g-1} \frac{\gamma^2}{\Phi_j^{\mathrm{Id}}(1 - 4\gamma^2)} + \frac{4d}{\gamma} + \sum_{j=0}^{g-1} \Phi_j^{\mathrm{Id}} \ge \frac{8d}{\gamma}\right)$$

$$\overset{(ii)}{\le} \frac{\mathbb{E}\left[\sum_{j=0}^{g-1} \frac{\gamma^2}{\Phi_j^{\mathrm{Id}}(1-4\gamma^2)} + \sum_{j=0}^{g-1} \Phi_j^{\mathrm{Id}}\right]}{4d/\gamma}$$

$$\overset{(iii)}{\le} \frac{g\mathbb{E}[\Phi_0^{\mathrm{Id}}] + \frac{\gamma^2}{1-4\gamma^2}\mathbb{E}\left[\sum_{j=0}^{g-1} \frac{1}{\Phi_j^{\mathrm{Id}}}\right]}{4d/\gamma}$$

$$\overset{(iv)}{\le} \frac{g\gamma + \frac{2\gamma g}{1-4\gamma^2}}{4d/\gamma}$$

where $(i)$ comes form the stopping criterion and the update rules for $u_j$ and $l_j$, $(ii)$ is Markov's inequality, $(iii)$ follows by Lemma 8, and $(iv)$ from Lemma 7. Using the fact that $\gamma < \frac{1}{4}$,

$$\Pr\left(m < g\right) \le \frac{g\gamma^2}{d} = p \tag{47}$$

$\square$

Next we define the "good" event $\Gamma$ that $u_m$ is indeed $\Omega(d/\gamma^2)$. Note that $\Gamma$ occurs with constant probability using Lemma 2.

**Definition 4.** *Define $\Gamma$ as the event that $\{u_m \ge \frac{d}{64\gamma^2}\}$. From Lemma 2, $\Pr(\Gamma) \ge \frac{3}{4}$.*

From the stopping criterion of the algorithm, we know that $u_j - l_j \le 8d/\gamma$, for $j < m$. We show that even for $j = m$ this inequality is true with a larger choice of constant.

**Lemma 9.** *For $\gamma < 1$, $u_m - l_m \le 9d/\gamma$.*

*Proof.* From Lemma 7, we know $\phi_{m-1}^{\mathrm{Id}} \ge \gamma/2$. Which implies $\gamma/\phi_{m-1}^{\mathrm{Id}} \le 2$. By the stopping criterion of the algorithm, $u_{m-1} - l_{m-1} < 8d/\gamma$. Using these two,

$$u_m - l_m = u_{m-1} - l_{m-1} + \frac{\gamma}{\phi_{m-1}^{\mathrm{Id}}}\left(\frac{1}{1-2\gamma} - \frac{1}{1+2\gamma}\right)$$

$$\le 8d/\gamma + 2\left(\frac{1}{1-2\gamma} - \frac{1}{1+2\gamma}\right)$$

$$\le 9d/\gamma$$

$\square$

Next we show that under the event $\Gamma$, the matrix $A_m$ is PSD, which is crucial towards bounding its condition number.

**Lemma 10.** *For $\gamma \leq \frac{1}{300}$, if the event $\Gamma$ (defined in Definition 4) occurs, $l_m > 0$.*

*Proof.* First observe that,

$$u_m = u_0 + \frac{\gamma}{(1 - 2\gamma)} \sum_{j=0}^{m-1} \frac{1}{\Phi_j^{\text{Id}}} \tag{48}$$

$$\implies \sum_{j=0}^{m-1} \frac{1}{\Phi_j^{\text{Id}}} = \left(u_m - \frac{2d}{\gamma}\right)\left(\frac{1 - 2\gamma}{\gamma}\right) \tag{49}$$

$$\implies l_m = \frac{-2d}{\gamma} + \frac{\gamma}{1 + 2\gamma}\left(u_m - \frac{2d}{\gamma}\right)\left(\frac{1 - 2\gamma}{\gamma}\right) \tag{50}$$

Thus if $u_m$ is large enough, the RHS will be $> 0$. It suffices to assume $\gamma \leq \frac{1}{300}$ for this statement to be true since conditioned on $\Gamma$, $u_m \geq \frac{d}{64\gamma^2}$. $\qquad\square$

Finally, conditioned on the event $\Gamma$ and invoking Lemma 9, we bound the condition number of $A_m$.

**Lemma 11.** *Conditioned on the event $\Gamma$ (defined in Definition 4), Algorithm 2's last iteration matrix $A_m$ has condition number $\frac{\lambda_{\max}(A_m)}{\lambda_{\min}(A_m)} \leq \frac{u_m}{l_m} \leq 1 + 3456\gamma$, for $\gamma \leq \frac{1}{700}$.*

*Proof.* From Lemma 17, the condition number of $A_m$ is at most,

$$\frac{u_m}{l_m} = \left(1 - \frac{u_m - l_m}{u_m}\right)^{-1} \tag{51}$$

Hence, it suffices to prove that $(u_m - l_m)/u_m$ is $\leq c\gamma$ with constant probability, assuming that $c\gamma \leq \frac{5}{6}$. We know from Lemma 9, $u_m - l_m \leq \frac{9d}{\gamma}$. Hence, it suffices to show that under the event $\Gamma$,

$$\frac{9d/\gamma}{u_k} \leq c\gamma \iff \frac{9d}{c\gamma^2} \leq u_m \tag{52}$$

Conditioned on the event $\Gamma$, $u_m \geq \frac{d}{64\gamma^2}$. Therefore, it suffices to choose $c \geq 576$. As $\gamma \leq \frac{1}{700}, c\gamma \leq \frac{5}{6}$. Finally,

$$\left(1 - \frac{u_m - l_m}{u_m}\right)^{-1} \leq 1 + \frac{c\gamma}{1 - c\gamma} \leq 1 + 3456\gamma. \tag{53}$$

$\qquad\square$

Lemma 11 directly translates to an upper bound on the eigenvalues of $Z^T Z$ which are nothing but the eigenvalues of $A_m$ up to a scaling factor of $(u_m + l_m)/2$.

**Lemma 12.** *Conditioned on the event $\Gamma$ (defined in Definition 4), $(1 - 1728\gamma)I \preceq Z^T Z \preceq (1 + 1728\gamma)I$.*

*Proof.* From Lemma 11, $\frac{u_m}{l_m} \leq 1 + 3456\gamma$.

$$Z^T Z = \frac{1}{\text{mid}} \sum_{j=1}^{m} w_j' U(x_j) U(x_j)^T = \frac{A_m}{\text{mid}}$$

Therefore, $\lambda(Z^T Z) \in \left[\frac{l_m}{\text{mid}}, \frac{u_m}{\text{mid}}\right]$. Also, given $\frac{u_m}{l_m} \leq 1 + 3456\gamma$, we have

$$\frac{u_m + l_m}{l_m} \leq 2 + 3456\gamma \tag{54}$$

$$\implies \frac{2}{2 + 3456\gamma} \leq \frac{l_m}{\frac{u_m + l_m}{2}} \tag{55}$$

$$\implies 1 - 1728\gamma \leq \frac{l_m}{\text{mid}} \tag{56}$$

A similar approach can be used to prove that $\frac{u_m}{\text{mid}} \leq 1 + 1728\gamma$. $\qquad\square$

This completes the proof in showing that Algorithm 2 satisfies the first property of being an $\epsilon$-well-balanced sampling procedure. Next, we prove that Algorithm 2 satisfies the second property ($\sum_{j=0}^{m-1} \alpha_j = O(1)$ and $\alpha_j K_{D_j} = O(\epsilon)$) which will complete the proof of Theorem 4 which we restate below.

**Theorem 4.** *Algorithm 2 is an $\epsilon$-well-balanced sampling procedure, where $0 < \epsilon < 1$. Furthermore, Algorithm 2 samples the labels of at most $O\left(\frac{R_X}{\epsilon}\right)$ points in $X_{un}$.*

*Proof.* To show that Algorithm 2 is an $\epsilon$-well-balanced sampling procedure, recall from the definition that we must show that with probability $\geq \frac{3}{4}$,

1. $\frac{3}{4} I \preceq Z^T Z \preceq \frac{5}{4} I$

2. $\sum_{j=0}^{m-1} \alpha_j = O(1)$ and for all $j = 0, \cdots, m-1$, $\alpha_j K_{D_j} = O(\epsilon)$.

Conditioned on the event $\Gamma$ which holds with probability $\geq \frac{3}{4}$, we show that both of these properties hold.

From Lemma 12, we have that $(1 - 1728\gamma)I \preceq Z^T Z \preceq (1 + 1728\gamma)I$. With $\gamma = \sqrt{\epsilon}/C_0$ with $\epsilon < 1$ and sufficiently large $C_0 > 0$, this implies that $\frac{3}{4} I \preceq Z^T Z \preceq \frac{5}{4} I$ which proves the first part.

On the other hand, to bound $\sum_{j=0}^{m-1} \alpha_j$, observe that

$$\sum_{j=0}^{m-1} \alpha_j = \sum_{j=0}^{m-1} \frac{\gamma}{\phi_j^{\text{Id}}} \cdot \frac{1}{\text{mid}} \leq \sum_{j=0}^{m-1} \frac{4}{\frac{u_m + l_m}{2}} \leq \frac{2d}{\gamma^2} \frac{8}{u_m} \leq 1024$$

where we use the fact that $u_m \geq \frac{d}{64\gamma^2}$ conditioned on $\Gamma$. Following the proof of Chen and Price [6, Lemma 5.1], we bound $\alpha_j K_{D_j}$ as follows:

$$\alpha_j K_{D_j} = \frac{\gamma}{\text{mid}} \cdot \frac{u_j - l_j}{2} = \gamma \frac{u_j - l_j}{u_m + l_m} \leq 512\gamma^2 = \frac{512\epsilon}{C_0^2}$$

where we upper bound $u_j - l_j \leq \frac{8d}{\gamma}$ using the stopping criterion of Algorithm 2, and lower bound $u_m + l_m \geq u_m \geq d/64\gamma^2$ conditioned on the event $\Gamma$. We also substitute $\gamma = \frac{\sqrt{\epsilon}}{C_0}$ and choose $C_0$ appropriately. $\qquad \square$

## A.5   Proof of lower bound (Theorem 5)

**Theorem 5.** *For any $0 < \epsilon < 1$, $d$ and any $\lambda > 0$. Suppose $X$ satisfies the condition, $\sigma_{\min}\left(X(X^T X)^{-1} X_{lab}^T\right) \geq \frac{\epsilon}{1-\epsilon}$. In the inductive setting, for each $X$ and learner there exists an instance of SSAR where if $\widehat{\beta}$ returned by the learner satisfies $\mathbb{E}\left[\|X\widehat{\beta} - Y\|_2^2\right] \leq (1 + \epsilon) \min_{\beta \in \mathbb{R}^d} \mathbb{E}\left[\|X\beta - Y\|_2^2\right]$ must query the labels of at least $\Omega(\frac{R_X}{\epsilon})$ points.*

Define $\begin{bmatrix} Y_{un} \\ Y_{lab} \end{bmatrix} = \begin{bmatrix} X_{un}\beta^* \\ 0 \end{bmatrix} + Z$. Here $Z$ is a noise vector where for each $x \in X$, the variance of $Z$ is $\nu_x^2$. We will assume that for $x \in X_{lab}$, $\nu_x^2 = 0$, while for the unlabeled points $x \in X_{un}$, we defer the specific choice of $\nu_x^2$ to later. Suppose the algorithm samples $(x_1, x_2, \cdots, x_T)$ and correspondingly observes labels $(y_1, \cdots, y_T)$. The underlying true parameter $\beta^*$ is assumed to be sampled from a distribution, defined below. Let $X = U\Sigma V^T$ be the SVD of $X$. Consider a vector $\alpha^*$ sampled uniformly from the vertices of a hypercuboid. Namely, $\alpha^* \sim \text{Unif}(\{\pm\kappa_1, \cdots, \pm\kappa_d\})$. Given $\alpha^*$, the ground truth parameter $\beta^*$ is defined as $V\alpha^*$.

Consider the parameter returned by the learner $\widehat{\beta}$. In the distributional setting, recall that the noise is sampled freshly while evaluating the generalization loss. The ground truth optimizer is therefore

$\widetilde{\beta} = (X^T X)^{-1}(X_{\text{lab}}^T X_{\text{lab}})\beta^* \triangleq \widetilde{\beta}$. The generalization loss itself can be computed as,

$$= \mathbb{E}_{P_{\beta^*}}\left[\left\|X(\widehat{\beta} - \widetilde{\beta})\right\|_2^2\right] + \left\|\begin{bmatrix} X_{\text{lab}} \\ X_{\text{un}} \end{bmatrix}\widetilde{\beta} - \begin{bmatrix} X_{\text{lab}}\beta^* \\ 0 \end{bmatrix}\right\|_2^2 + \sum_{x \in X_{\text{lab}}} \nu_x^2 \tag{57}$$

$$= \mathsf{OPT} + \mathbb{E}_{P_{\beta^*}}\left[\left\|X(\widehat{\beta} - \widetilde{\beta})\right\|_2^2\right] \tag{58}$$

$$= \mathsf{OPT} + \mathbb{E}_{P_{\beta^*}}\left[\left\|X\left(\widehat{\beta} - (X^T X)^{-1}(X_{\text{lab}}^T X_{\text{lab}})\beta^*\right)\right\|_2^2\right] \tag{59}$$

Assume that $X_{\text{lab}}$ is a tall and full-rank matrix. Let $U_{\text{lab}}\Sigma_{\text{lab}}V_{\text{lab}}^T$ be the SVD of $X_{\text{lab}}$. Define $W = V_{\text{lab}}\Sigma_{\text{lab}}V_{\text{lab}}^T$, which is invertible by the full-rankness assumption on $X_{\text{lab}}$. Then $(X_{\text{lab}}^T X_{\text{lab}}) = V_{\text{lab}}\Sigma_{\text{lab}}^T\Sigma_{\text{lab}}V_{\text{lab}}^T$. Next, consider the matrix $X(X^T X)^{-1}W^T$, and let its SVD be $U\Sigma V^T$.

From eq. (59), the generalization loss can be rewritten as,

$$= \mathsf{OPT} + \mathbb{E}_{P_{\beta^*}}\left[\left\|X\left(\widehat{\beta} - (X^T X)^{-1}(W^T W)\beta^*\right)\right\|_2^2\right] \tag{60}$$

$$= \mathsf{OPT} + \mathbb{E}_{P_{\beta^*}}\left[\left\|X(X^T X)^{-1}W^T\left((W^T)^{-1}(X^T X)\widehat{\beta} - W\beta^*\right)\right\|_2^2\right] \tag{61}$$

Consider the matrix $X(X^T X)^{-1}W^T$. Let its SVD be $U\Sigma V^T$. Define $\widehat{\alpha} = V^T(W^T)^{-1}(X^T X)\widehat{\beta}$ and $\alpha^* = V^T W\beta^*$. Then, the generalization loss is,

$$= \mathsf{OPT} + \mathbb{E}_{P_{\beta^*}}\left[\|\Sigma(\widehat{\alpha} - \alpha^*)\|_2^2\right], \tag{62}$$

Given an algorithm which returns a $\widehat{\beta}$ (and equivalently a $\widehat{\alpha}$), it induces a test, which returns the vector in $\mathcal{A} \triangleq \{\pm\kappa_i\}_{i=1}^d$ closest in $L_2$ norm to $\alpha^*$. If the test makes a mistake in some coordinates $S$ of $\alpha^*$, the additive error induced is,

$$\sum_{i \in S} \kappa_i^2(\Sigma_{ii})^2. \tag{63}$$

With this viewpoint, suppose $\alpha^*$, which can span $\mathbb{R}^d$ is distributed uniformly over $\mathcal{A}$. Then, the error made by the learner is lower bounded by error incurred by the best testing algorithm for each coordinate. Namely,

$$\sup_{\alpha^* \in \mathcal{A}} \mathbb{E}_{P_{\beta^*}}\left[\|\Sigma(\widehat{\alpha} - \alpha^*)\|_2^2\right] \tag{64}$$

$$\geq \inf_{\Psi_1, \cdots, \Psi_d} \mathbb{E}_{\alpha^* \sim \text{Unif}(\mathcal{A})}\left[\sum_{i=1}^d \kappa_i^2(\Sigma_{ii})^2 \mathbb{E}_{P_{\alpha^*}}\left[\mathbb{I}(\Psi_i(\text{adaptive data}) \neq \alpha_i^* \in \{\pm 1\})\right]\right] \tag{65}$$

$$\geq \mathbb{E}_{\alpha^* \sim \text{Unif}(\mathcal{A})}\left[\sum_{i=1}^d \kappa_i^2(\Sigma_{ii})^2\left(1 - \mathsf{TV}(P_{\alpha^*}, P_{\overline{\alpha}_i^*})\right)\right] \tag{66}$$

Note that the learner has access to (noisy) labelled samples from $X_2$; thus $P_{\beta^*}$ denotes the joint distribution over $(X_{\text{lab}}, Y_{\text{lab}}, X_1, Y_1, \cdots, X_T, Y_T)$ induced by the algorithm (and where $X_1, Y_1, \cdots, X_T, Y_T$ are generated in a Markovian fashion). Also note that $\overline{\alpha}_i^*$ is the parameter obtained by flipping the $i^{th}$ coordinate of $\alpha^*$. Observe that,

$$\mathbb{E}_{\alpha^* \sim \text{Unif}(\mathcal{A})}\left[\sum_{i=1}^d \kappa_i^2(\Sigma_{ii})^2 \mathsf{TV}(P_{\alpha^*}, P_{\overline{\alpha}_i^*})\right] \tag{67}$$

$$\leq \mathbb{E}_{\alpha^* \sim \text{Unif}(\mathcal{A})}\left[\sum_{i=1}^d \kappa_i^2(\Sigma_{ii})^2\sqrt{2\mathsf{KL}(P_{\alpha^*}, P_{\overline{\alpha}_i^*})}\right] \tag{68}$$

$$\leq \sqrt{2}\mathbb{E}_{\alpha^* \sim \text{Unif}(\mathcal{A})}\left[\sqrt{\sum_{i=1}^d \gamma_i^2\left(\kappa_i^2(\Sigma_{ii})^2\right)^2}\sqrt{\sum_{i=1}^d \frac{1}{\gamma_i^2}\mathsf{KL}(P_{\alpha^*}, P_{\overline{\alpha}_i^*})}\right]. \tag{69}$$

Where the last step follows by Cauchy Schwarz inequality. Next, observe that,

$$\mathsf{KL}(P_{\alpha^*}, P_{\overline{\alpha}_i^*}) \tag{70}$$

$$= \mathbb{E}_{P_{\beta^*}}\left[\log\left(\frac{P_{\beta^*}}{P_{\overline{\beta}_i^*}}\right)\right] \tag{71}$$

$$= \frac{1}{2}\mathbb{E}_{P_{\beta^*}}\left[\frac{\sum_{t=1}^T (Y_t - \langle X_t, \overline{\beta}_i^*\rangle)^2 - (Y_t - \langle X_t, \beta^*\rangle)^2}{\nu_{X_t}^2} + \frac{\sum_{(x,y)\in X_{\text{lab}}\times Y_{\text{lab}}} (y)^2 - (y)^2}{\nu_x^2}\right] \tag{72}$$

$$= \frac{1}{2}\sum_{t=1}^T \mathbb{E}_{P_{\beta^*}}\left[\frac{2B_t\left(\langle X_t, \beta^*\rangle - \langle X_t, \overline{\beta}_i^*\rangle\right) - \left(\langle X_t, \beta^*\rangle^2 - \langle X_t, \overline{\beta}_i^*\rangle^2\right)}{\nu_{X_t}^2}\right] \tag{73}$$

$$= \frac{1}{2}\sum_{t=1}^T \mathbb{E}_{P_{\beta^*}}\left[\frac{\left\langle X_t, \beta^* - \overline{\beta}_i^*\right\rangle^2}{\nu_{X_t}^2}\right] \tag{74}$$

Note that $\beta^* = W^{-1}V\alpha^*$ where $\alpha^* \in \{\pm\kappa_i\}_{i=1}^d$ and likewise, $\overline{\beta}_i^* = W^{-1}V\overline{\alpha}_i^*$. Therefore, $\beta^* - \overline{\beta}_i^* = 2\kappa_i W^{-1}Ve_i$. Overall, we get,

$$\sum_{i=1}^d \frac{1}{\gamma_i^2}\mathsf{KL}(P_{\alpha^*}, P_{\overline{\alpha}_i^*}) = \frac{1}{2}\mathbb{E}_{P_{\beta^*}}\left[\sum_{t=1}^T\sum_{i=1}^d \frac{1}{\gamma_i^2}\frac{\left\langle X_t, \beta^* - \overline{\beta}_i^*\right\rangle^2}{\nu_{X_t}^2}\right] \tag{75}$$

$$= 2\mathbb{E}_{P_{\beta^*}}\left[\sum_{t=1}^T\sum_{i=1}^d \frac{\kappa_i^2}{\gamma_i^2\nu_{X_t}^2}\left\langle X_t, W^{-1}Ve_i\right\rangle^2\right] \tag{76}$$

Choosing $\nu_x^2 = \frac{1}{\epsilon}\sum_{i=1}^d \frac{\kappa_i^2}{\gamma_i^2}\langle x, W^{-1}Ve_i\rangle^2$ for $x \in X$, we get that the RHS is equal to $2\epsilon T$. The overall error of the learner is,

$$\mathbb{E}_{\alpha^*\sim\text{Unif}(\mathcal{A})}\left[\sum_{i=1}^d \kappa_i^2(\Sigma_{ii})^2\left(1 - \mathsf{TV}(P_{\alpha^*}, P_{\overline{\alpha}_i^*})\right)\right] \tag{77}$$

$$\geq \sum_{i=1}^d \kappa_i^2(\Sigma_{ii})^2 - 2\sqrt{\sum_{i=1}^d \gamma_i^2\left(\kappa_i^2(\Sigma_{ii})^2\right)^2}\sqrt{\epsilon T} \tag{78}$$

Next we deal with computing $\mathsf{OPT}$. First we introduce a new notation, $\Delta_i$ and an associated lemma.

**Lemma 13.** *For $i \in [d]$, define*

$$\Delta_i = e_i^T V^T W (X^T X)^{-1} W^T V e_i \tag{79}$$

*Then, $\sum_{i=1}^d \Delta_i = \mathsf{R}_X$.*

*Proof.* By direct calculation,

$$\sum_{i=1}^d \Delta_i = \mathsf{Tr}\left(V^T W (X^T X)^{-1} W^T V\right) \tag{80}$$

$$= \mathsf{Tr}\left((X^T X)^{-1} W^T W\right) \tag{81}$$

$$= \mathsf{Tr}\left((X^T X)^{-1}(X_{\text{lab}}^T X_{\text{lab}})\right) = \mathsf{R}_X, \tag{82}$$

where the middle equation follow by commutativity of the trace operator and the fact that $V$ is an rotation matrix. $\qquad\square$

**Lemma 14.** $\mathsf{OPT} \leq \frac{1}{\epsilon}\sum_{i=1}^d \frac{\kappa_i^2}{\gamma_i^2} + 2\sum_{i=1}^d (\Sigma_{ii}^2 + 1)\kappa_i^2.$

*Proof.* Note that, $\mathsf{OPT} = \sum_{x \in X_{\mathrm{lab}}} \nu_x^2 + \left\| X\widetilde{\beta} - \begin{bmatrix} X_{\mathrm{lab}}\beta^* \\ 0 \end{bmatrix} \right\|_2^2$. Focusing on the second term,

$$\left\| X\widetilde{\beta} - \begin{bmatrix} X_{\mathrm{lab}}\beta^* \\ 0 \end{bmatrix} \right\|_2^2 \leq 2 \left\| X\widetilde{\beta} \right\|_2^2 + 2 \left\| X_{\mathrm{lab}}\beta^* \right\|_2^2 \tag{83}$$

$$= 2 \left\| X(X^TX)^{-1}(W^TW)\beta^* \right\|_2^2 + 2 \left\| X_{\mathrm{lab}}W^{-1}V\alpha^* \right\|_2^2 \tag{84}$$

$$\leq 2 \left\| U\Sigma V^T W\beta^* \right\|_2^2 + 2 \left\| U_{\mathrm{lab}}V_{\mathrm{lab}}^T V\alpha^* \right\|_2^2 \tag{85}$$

$$= 2 \left\| \Sigma\alpha^* \right\|_2^2 + \left\| \alpha^* \right\|_2^2 \tag{86}$$

$$= 2 \sum_{i=1}^d (1 + \Sigma_{ii}^2)\kappa_i^2 \tag{87}$$

Where $\beta^* = W^{-1}V\alpha^*$. On the other hand, the sum of variances can be upper bounded as,

$$\sum_{x \in X} \nu_x^2 = \sum_{x \in X_{\mathrm{lab}}} \nu_x^2 = \frac{1}{\epsilon} \sum_{i=1}^d \sum_{x \in X_{\mathrm{lab}}} \frac{\kappa_i^2}{\gamma_i^2} \langle x, W^{-1}Ve_i \rangle^2 \tag{88}$$

$$= \frac{1}{\epsilon} \sum_{i=1}^d \frac{\kappa_i^2}{\gamma_i^2} \| XW^{-1}Ve_i \|_2^2 \tag{89}$$

$$= \frac{1}{\epsilon} \sum_{i=1}^d \frac{\kappa_i^2}{\gamma_i^2}, \tag{90}$$

where the last equation uses the fact that $X_{\mathrm{lab}}^T X_{\mathrm{lab}} = W^TW$, so $(W^T)^{-1}(W^TW)W^{-1} = I$, and therefore $\| X_{\mathrm{lab}}W^{-1}Ve_i \|_2^2 = 1$. $\qquad\square$

We next describe the choice of $\kappa_i$'s and $\gamma_i$'s. Define,

$$\kappa_i^2 = \frac{\Delta_i}{\Sigma_{ii}^2} \text{ and } \gamma_i^2 = \frac{1}{\Sigma_{ii}^2}. \tag{91}$$

By the above choices, as a corollary of Lemma 14,

**Corollary 2.** *Assume that $\Sigma_{ii}^2 \geq \frac{\epsilon}{1-\epsilon}$ for all $i \in [d]$. Then, $\mathsf{OPT} \leq \frac{3\mathrm{R}_X}{\epsilon}$.*

*Proof.* From Lemma 14,

$$\mathsf{OPT} \leq \frac{1}{\epsilon} \sum_{i=1}^d \frac{\kappa_i^2}{\gamma_i^2} + 2 \sum_{i=1}^d (\Sigma_{ii}^2 + 1)\kappa_i^2 \tag{92}$$

$$\leq \frac{1}{\epsilon} \sum_{i=1}^d \Delta_i + 2 \sum_{i=1}^d (\Sigma_{ii}^2 + 1)\frac{\Delta_i}{\Sigma_{ii}^2} \tag{93}$$

$$\overset{(i)}{\leq} \frac{\mathrm{R}_X}{\epsilon} + \frac{2\mathrm{R}_X}{\epsilon} \tag{94}$$

where $(i)$ follows from Lemma 13 for the first term, and assumes that $\Sigma_{ii}^2 \geq \frac{\epsilon}{1-\epsilon}$ for all $i \in [d]$ for the second. $\qquad\square$

$$\sum_{i=1}^d \frac{\kappa_i^2}{\gamma_i^2} = \sum_{i=1}^d \kappa_i^2 \Sigma_{ii}^2 = \sum_{i=1}^d \Delta_i = \mathrm{R}_X. \tag{95}$$

where the last equation follows from Lemma 13.

Next we bound the generalization error of the learner.

**Lemma 15.** *The generalization error of the learner is lower bounded by $\mathrm{R}_X - 2\sqrt{\epsilon\mathrm{R}_X T}$.*

*Proof.* From eq. (78), the generalization error of any learner is lower bounded by,

$$\sum_{i=1}^{d} \kappa_i^2 (\Sigma_{ii})^2 - 2\sqrt{\sum_{i=1}^{d} \gamma_i^2 \left(\kappa_i^2 (\Sigma_{ii})^2\right)^2} \sqrt{\epsilon T} \tag{96}$$

$$\geq \sum_{i=1}^{d} \Delta_i - 2\sqrt{\sum_{i=1}^{d} \frac{\Delta_i^2}{\Sigma_{ii}^2}} \sqrt{\epsilon T} \tag{97}$$

Note that, $X(X^T X)^{-1} W^T = U\Sigma V^T$. Therefore, $V^T W (X^T X)^{-1} W^T V = \Sigma^T \Sigma$ and we have the equation, $\Delta_i = \Sigma_{ii}^2$ for all $i \in [d]$. Therefore, from eq. (97), the generalization error of any learner can be lower bounded by,

$$\geq \sum_{i=1}^{d} \Delta_i - 2\sqrt{\sum_{i=1}^{d} \Delta_i} \sqrt{\epsilon T} \tag{98}$$

Invoking Lemma 13 completes the proof. □

Therefore, from Corollary 2 and Lemma 15, under the assumption that $\Sigma_{ii}^2 \geq \frac{\epsilon}{1-\epsilon}$ for all $i \in [d]$, the approximation factor of any learner is lower bounded by

$$1 + \frac{R_X - 2\sqrt{\epsilon R_X T}}{3R_X/\epsilon} \tag{99}$$

If $T \leq \frac{R_X}{\epsilon}$ the approximation factor must be $1 + \Omega(\epsilon)$. This completes the proof.

## B   Auxiliary lemmas

**Lemma 16.** *For $\gamma \leq \frac{1}{4}$, $w_j w_j^T \preceq 2\gamma(A_j - l_{j+1}I)$.*

*Proof.* First observe that,

$$w_j w_j^T \preceq \gamma(A_j - l_j I) = \gamma(A_j - l_{j+1}I) + \gamma(l_{j+1} - l_j)I \tag{100}$$

Therefore, it suffices to show that $l_{j+1} - l_j \preceq (A_j - l_{j+1}I)$, or in other words, $l_{j+1} - l_j \leq \lambda_{\min}(A_j - l_{j+1}I)$ to complete the proof. By definition,

$$l_{j+1} - l_j = \frac{\gamma}{(1 - 2\gamma)\Phi_j^{\mathsf{ld}}} \leq \frac{\gamma}{1 - 2\gamma} \lambda_{\min}(A_j - l_j I) \leq \frac{1}{2}\lambda_{\min}(A_j - l_j I) \tag{101}$$

where the last inequality uses the fact that $\gamma \leq \frac{1}{4}$. Therefore, $2l_{j+1} - l_j \leq \lambda_{\min}(A_j)$ and $l_{j+1} - l_j \leq \lambda_{\min}(A_j - l_{j+1})$. Plugging this back into eq. (100), we arrive at the claim of the lemma. □

**Lemma 17.** *For $\gamma < 1$, in each iteration $j = 0, \cdots, m$ of Algorithm 2, the condition $l_j I \preceq A_j \preceq u_j I$ is satisfied.*

*Proof.* The proof follows by induction. For $j = 0$, $A_j = 0$ and trivially satisfies the condition $\frac{2d}{\gamma}I = -l_j I \preceq A_j \preceq u_j I = \frac{2d}{\gamma}I$. By the induction hypothesis, we assume that $l_j I \preceq A_j \preceq u_j I$ henceforth in the proof. For any point $x$, observe that,

$$p_x \Phi_j^{\mathsf{ld}} = U(x)^T \left((u_j I - A_j)^{-1} + (A_j - l_j I)^{-1}\right) U(x) \tag{102}$$

$$\geq U(x)^T (u_j I - A_j)^{-1} U(x) \tag{103}$$

Observe that for any vector $v$ and PSD matrix $B$, $vv^T \preceq (v^T B^{-1}v)B$. Therefore, for any point $x$,

$$U(x)U(x)^T \preceq (U(x)^T (u_j I - A_j)^{-1} U(x))(u_j I - A_j) \tag{104}$$

$$\overset{(i)}{\preceq} p_x \Phi_j^{\mathsf{ld}}(u_j I - A_j) \tag{105}$$

where $(i)$ uses eq. (103). Similarly by lower-bounding eq. (102) by $U(x)^T(A_j - l_j I)^{-1}U(x)$ and use a similar approach to prove that for any $x$,

$$U(x)U(x)^T \preceq p_x \Phi_j^{\text{Id}}(A_j - l_j I) \tag{106}$$

Choosing $x = x_j$ in eq. (105), as a special case,

$$A_{j+1} - A_j = \frac{\gamma}{p_j \Phi_j^{\text{Id}}}U(x_j)U(x_j)^T \preceq \gamma(u_j I - A_j) \tag{107}$$

Using the induction hypothesis, we use this to prove that $A_{j+1} \preceq u_{j+1}I$. Indeed, eq. (107) implies that,

$$(u_j I - A_j) - (u_j I - A_{j+1}) = A_{j+1} - A_j \preceq \gamma(u_j I - A_j) \tag{108}$$

Therefore,

$$(1 - \gamma)(u_j I - A_j) \preceq u_j I - A_{j+1} \preceq u_{j+1}I - A_{j+1} \tag{109}$$

And using the induction hypothesis that $u_j I - A_j \succeq 0$ completes the proof that $A_{j+1} \preceq u_{j+1}I$. On the other hand, to prove that $A_{j+1} \succeq l_{j+1}I$, summing eq. (106) over all $x$ and noting that $\sum_x U(x)U(x)^T = I$,

$$\frac{1}{\Phi_j^{\text{Id}}}I \preceq A_j - l_j I \tag{110}$$

Finally, observe that,

$$A_{j+1} - l_{j+1}I = (A_j - l_j I) + \left( \frac{\gamma}{\Phi_j^{\text{Id}}}\frac{1}{p_j}U(x_j)U(x_j)^T - \frac{\gamma}{1 + 2\gamma}\frac{1}{\Phi_j^{\text{Id}}}I \right) \tag{111}$$

$$\succeq (A_j - l_j I) - \frac{\gamma}{1 + 2\gamma}\frac{1}{\Phi_j^{\text{Id}}}I \tag{112}$$

$$\overset{(i)}{\succeq} \frac{1}{\Phi_j^{\text{Id}}}I - \frac{\gamma}{1 + 2\gamma}\frac{1}{\Phi_j^{\text{Id}}}I \tag{113}$$

$$\succeq 0 \tag{114}$$

where $(i)$ uses eq. (110). $\qquad\square$