# OpenReview forum: "Semi-supervised Active Linear Regression"
_NeurIPS.cc/2022/Conference — NeurIPS 2022 Accept_

### Official Review · Reviewer_jxU1 · 2022-07-11

**Rating:** 7
**Confidence:** 4
**Soundness:** 3 good
**Presentation:** 4 excellent
**Contribution:** 3 good

**Summary:**

This paper proposes to study a active semi-supervised least-squares regression problem, where some labelled points are already given and the goal is to minimise the least-squares loss using as few as possible queries to the unlabelled points.

Interestingly, the minimisation on the labelled points can be used to encode a regularisation term, hence, this problem generalises active ridge regression and even active kernel ridge regression.

The authors propose to use the instance-dependent parameter "reduced rank" $R_X$, improving previous bounds which only depend on the dimension $d$ of the instances, and give matching lower bounds in the specific case of active ridge regression.

They motivate it by stating that it determines the query complexity of often used leverage sampling as $O(R_X\log d / \epsilon)$ and state a near-matching lower bound. One of the main contributions is a polynomial-time algorithm based on previous results of [6] and [16] to remove the O(log d) term.

In common special cases, $R_X$ reduces to known parameters like the statistical dimension or the effective dimension.

**Questions:**

The lower bound in Theorem 2 seems to not exactly match the upper bound of Theorem 1, in particular, it misses a $1/\epsilon$. What is the reason for that? Shouldn't it be possible to improve the $1/4 R_X \log d$ lower bound with some additional $\epsilon$ term? In particular, for very small $\epsilon$ it seems natural that the lower bound should increase.

By the definition of $R_X$ (and in particular, its equivalence to the sum over $||U(x)||^2$), it seems that $R_X\leq d$. Please discuss this, as otherwise it is not immediately clear whether the achieved bounds using $R_X$ are always better than previous bounds using $d$.

While the semi-supervised setting is very interesting from a practical perspective, it is not clear whether this matters here. All bounds and proofs seem to be mostly concerned with the number of queries and do not take the size or structure of the labelled set into account. E.g., a worst-case could be that the whole matrix $X_{lab}$ consists of the same single instances multiple times, which would provide barely additional information compared to just the active learner (without an additional labelled set). Please comment here.

**Limitations:**

The authors focus on the distribution-free transductive case where they bound the error only on the given points $X_{lab}$ and $X_{un}$. Are statistical generalisation bounds possible, i.e., if all $x$ are drawn iid from some (unknonw) distribution, are error guarantees for an unseen $x$ possible?

**Strengths And Weaknesses:**

This paper is well-written and tackles an interesting problem. It will probably be appreciated by a theoretical subcommunity of NeurIPS.

The authors provide a strong general bound using the instance-dependent parameter $R_X$, which strengthens previous bounds in known special cases. They show a probabilistic lower bound matching their query upper bound, showing that it is asymptotically optimal.

Please discuss previous and proposed lower bounds more thoroughly. Without checking [6] it is unclear whether $O(d/\epsilon)$ is optimal, in the sense that there are instances requiring $\Omega(d/\epsilon)$. It is mentioned in line 103 but it's not completely clear that a matching lower bound exists. Maybe an overview table with the different bounds in the different settings helps to get a quick overview?

Minor points:
* "costly' " in the abstract
* line 39: union notation for matrices is not common. They are just stacked row-wise, as in equation (4)?
* Definition 1 seems convoluted.
* Please formally define "$\preccurlyeq$" (Loewner order?)
* This paper uses the term "agnostic" in a not common sense. In the ML and ML theory literature, agnostic usually refers to the case where we want to PAC learn using an iid training sample and a given hypothesis space, yet, the instances might be labelled arbitrary (i.e., agnostic = not-realizable=ground truth is not necessarily from the hypothesis space), see e.g., Understanding Machine Learning (Shai Shalev-Shwartz and Shai Ben-David, 2014). The authors seem to call the fact that they do not use iid sampling assumptions but rather have worst-case bounds on a given set of points (which do not have to be sampled) "agnostic".

Potential typos:
* What is $X_1$ in line 170. It seems to be $X_{un}$.
* Weird gaps after "Alg" in line 283 and 284.
* $O(..)$ missing around $R_X/\epsilon$ in line 351.
* line 196: "which decides which points to query the labels of points" sounds not correct.

---

> ### Author Response · Authors · 2022-08-02
> **Response to Reviewer jxU1**
>
> We thank the reviewer for the detailed feedback and constructive criticism.
>
> We shall fix the typographic errors in the subsequent version of the paper.
> We shall also improve the discussion surrounding the reference [6] with regards to the discussion of the lower bound of $\Omega(d/\epsilon)$ for active regression. Note in general the lower bound we propose in Theorem 5 of the paper shows instance dependent optimality, which is stronger than the worst-case optimality result in the results of [6]. Moreover, in the worst case where the labeled dataset is empty, the reduced rank evaluates to $d$, and thus our lower bound is able to also recover the lower bound in [6]. We shall include a more comprehensive discussion and a table in the paper, if space permits.
>
> Furthermore, we point out to the reviewer that we use the term "agnostic" in the same sense as the reviewer understands - the labels come from a function which is not necessarily in the learner's hypothesis/regression function class. We use the term active learning to indicate the absence of the iid sampling assumption (the learner can interact with a labeler to query the label of any input point).
>
> In the lower bound of Theorem 2, we only intend to show that for some range of parameters, leverage score sampling pays an additional $\log(d)$ factor. We believe that the proof of this result can be extended to an arbitrary range of $\epsilon$, by tweaking the underlying instance to not be symmetrical along all basis vectors. The key idea is to note that when certain directions have fewer points along those directions, leverage score sampling would not tend to sample them as much, and therefore, the learner is forced to sample more points in order to recover the same error guarantees as before. This results in an additional $1/\epsilon$ factor.
>
> As we point out in the response to Reviewer qdqB, the reduced rank is always upper bounded by $d$, which is achieved when the labeled dataset is empty. In all other cases, the reduced rank is at most $d$, and can be significantly smaller if the labeled and unlabeled datasets are "aligned" in the sense of having similar covariance matrices up to scaling factors.
>
> Finally, we note that the motivation of designing instance dependent guarantees that depend on the some problem dependent parameter (here, the reduced rank) is exactly the motivation of our paper. Yes, in the worst case, the reduced rank can be as large as $d$ when the labeled and unlabeled datasets are not aligned at all (or the labeled dataset is empty). But practical instances are not likely to have these pathologies as one would expect some level of alignment between the labeled and unlabeled dataset covariances. In the case that the reviewer points out where the same example is repeated many times, no algorithm can make use of the labeled dataset to improve the performance. This is reflected in the reduced rank for these kinds of instances being as large as $d$. Thus, our bounds should be interpreted as providing improvement over the worst-case when the underlying problem instances are not pathological.
>
> Finally, as we note in the response to Reviewer MhgG, the results in our paper can directly be extended to the case of generalization bounds where the inputs comes from a distribution. Here, $X_{\text{un}}$ grows to be infinitely large and each point is scaled by a tiny scaling factor, which captures the weight of the distribution on that input. As before, the learner can choose to sample a small subset of the points (say in X_{\text{lab}}$ and observes the corresponding labels. The quantities being bounded in the main result, Theorem 3 exactly becomes the generalization loss.

---

> > ### Comment · Reviewer_jxU1 · 2022-08-03
> > **Thanks for clarification**
> >
> > Thanks for the clarification and comments! I am now even more convinced that this is a good paper.

---

### Official Review · Reviewer_MhgG · 2022-07-12

**Rating:** 7
**Confidence:** 3
**Soundness:** 4 excellent
**Presentation:** 4 excellent
**Contribution:** 3 good

**Summary:**

This work tackles the problem of semi-supervised active linear regression (SSALR), which generalizes active ridge regression and active kernel ridge regression. The main focus of the paper is to establish instance dependent results on SSALR - in particular, matching upper and lower bounds -, which is done by using the novel notion of reduced rank. In the active ridge regression and active kernel ridge regression this new instance dependent parameter generalizes the key quantities statistical dimension and effective dimension, respectively.


**Questions:**

- It seems like one connects distributional SSALR to SSALR by stacking an infinite amount of copies of the original dataset with perturbed labels. If so, does Algorithm 1 still apply as is? It seems like it involves an SVD computation.
- Are these algorithms implementable in practice?


**Limitations:**

The theoretical limitations are adequately addressed. Although this work is theoretical, it might still be valuable to mention what could go wrong if the suggested algorithms were actually deployed.


**Strengths And Weaknesses:**

Strengths:
- This problem nicely generalizes well-known frameworks.
- The paper is clearly presented and technically sound.

Weaknesses:
- No experimental results limit the impact of the work.
- It is surprising that the absence of potential negative societal impacts is not justified.

---

> ### Author Response · Authors · 2022-08-02
> **Response to Reviewer MhgG**
>
> We thank the reviewer for their feedback and questions.
>
> We shall include a discussion of the potential negative impacts of the paper in its subsequent version. Addressing some of the questions raised in the review,
>
> The connection between distributional SSALR and SSALR requires stacking an infinite set of copies of the original dataset (and downweighting each copy). The SVD of this matrix can be computed by computing the SVD where there is just a single copy weighted by $1$. Thus the algorithm can therefore be implemented efficiently even in the distribution version of the problem. See the response to Reviewer jxU1 for more details.
>
> In theory, the run time of the algorithm is polynomial in the dimension of the problem and the approximation error $\epsilon$ involving matrix-vector products, matrix inversion and SVDs. However, the current version of the algorithm does not run in near-linear time. We mention that the theoretical time complexity of the algorithm can be further improved by only periodically recomputing the sampling probabilities p_x as is done in the original paper [1] which introduces the randomized BSS algorithm. However we focus on getting optimal statistical error guarantees in our paper, and leave the design of near-optimal time algorithms for future work.
>
> [1]: YT Lee and H. Sun, “Constructing Linear-Sized Spectral Sparsification in Almost-Linear Time”, Arxiv: 1508.03261

---

### Official Review · Reviewer_qdqB · 2022-07-15

**Rating:** 6
**Confidence:** 3
**Soundness:** 3 good
**Presentation:** 2 fair
**Contribution:** 3 good

**Summary:**

Summary:
The paper addresses the problem of semi-supervised active regression and gives algorithms with instance-optimal query complexity for this problem. Their query complexity results are based on novel notions of complexity parameters introduced in the paper. In particular, they propose an instance dependent parameter, called reduced rank $R_X$. It measures the relative importance of labeled points against the unlabeled ones. This quantity $R_X$ is shown to be equal to statistical dimension $sd_{\lambda} $ (in the active ridge regression setup) and effective dimension $d_{\lambda } $ in the active kernel ridge regression. Thus, leading to query complexity results of $O(sd_{\lambda}/\epsilon$ and $ O(d_{\lambda}/\epsilon) $ for active ridge regression and active kernel ridge regression setups.  Their algorithm is based on randomized BSS spectral sparsification algorithm from [16,4]. It assigns weights on the dataset and depending on these weights a subset of unlabeled data is selected, and its labels are queried. Designing a sampling algorithm which is guaranteed to query labels only for a limited number of unlabeled points is challenging and this paper gives an algorithm to do this.


**Questions:**

1. Could you please include interpretation of the proposed quantities in early in the paper and give some interpretation of the bounds?
2. It might be brief, but its good to see a comparison with the query complexity of passive sampling for this problem.

**Limitations:**

Limitations are discussed briefly in section 6. The algorithm is based on SVD of the instance matrix thus it is computationally expensive. They leave design of near-linear time algorithms as future work.


**Strengths And Weaknesses:**


Strengths:
1.	They formalize a notion of reduced rank $R_X$ that is an instance dependent parameter capturing problem complexity. They show that this parameter is equal to previously proposed complexity parameters $sd_{\lambda}$ and $d_{\lambda}$ for the active ridge regression and active kernel ridge regression. I think, it is a good abstraction and helps to connect and understand various problem settings.
2.	Previously known query complexity guarantees of $O(d/\epsilon)$ are not optimal for active ridge regression, in the sense that the dependence on $d$ is not ideal. The paper gives a query complexity result of $O(sd_{\lambda}/\epsilon)$ and shows that it is optimal in the worst case.
3.	They also improve the query-complexity of active kernel ridge regression from $O(d_{\lambda} log(N))$ to $d_{\lambda}/\epsilon$. Here N is the number of data points so the bound given in paper is free from the number of datapoints which could be large in practice.


Weaknesses:
1.	Guarantee is on the loss calculated on the X_un and X_lab. The problem setup is focused on 1+\epsilon factor approximation of the loss function using minimal number of labeled points. However, it doesn’t say much about the generalization error of \hat{\beta}.
2.	It lacks the interpretation of the quantities proposed in the paper, it might be better to give some intuitive understanding of R_X, sd_\lambda, d_\lambda and how should one expect them to behave, are they going to be small or could be very large, are they smaller than d?
3.	I didn’t see a comparison of query complexity with passive sampling. How do these complexities compare against random i.i.d sampling?

---

> ### Author Response · Authors · 2022-08-02
> **Response to Reviewer qdqB**
>
> We thank the reviewer for the constructive feedback and questions.
>
> First and foremost, while the statistical dimension $\text{sd}_\lambda$ and effective dimension $d_\lambda$ have been covered extensively in the literature of ridge regression, and kernel methods, in the interest of keeping things self contained, we shall improve the presentation of these quantities as well as the reduced rank by including more interpretations. In essence, the reduced rank itself captures the alignment of the labeled dataset with the unlabeled dataset - if both datasets have similar covariance matrices, then the reduced rank is automatically small. Likewise, the reduced rank can be shown to monotonically decrease as more datapoints are added to the labeled dataset (one may always choose to ignore these points). It is a short computation to therefore show that the reduced rank is always less than $d$ (i.e. when the labeled dataset is empty). We shall include this result as a lemma in the paper. By extension, in the worst case over all problem instances, the sample complexity with active sampling for the proposed algorithm is upper bounded by $d/\epsilon$.
> In contrast, in the i.i.d. setting the optimal sample complexity is known to be $\Theta(d/\epsilon^2)$, achieved by empirical risk minimization. Thus, the active setting offers an improvement of an $\epsilon$ factor in the sample complexity compared to passive sampling. In addition, the instance dependent (reduced rank based) bound improves the dependence on the dimension as well, when the labeled dataset and unlabeled dataset are more “aligned”.
>
> Finally, we would like to clarify that while we study the setting where the input points are not sampled from a distribution, the results also carry over to the setting where the inputs come from a distribution. Extending our main result to this setting, the resulting guarantee would be a bound on precisely the generalization loss (see the response to Reviewer jxU1 for more details). We shall include this as a remark in the paper, since this is an important point to make as the reviewer correctly identified.

---

> > ### Comment · Reviewer_qdqB · 2022-08-08
> > **Thanks and Further Comments**
> >
> > I would like to thank the authors for the response.
> >
> > I think it might be helpful to include a table in the first few pages to summarize and compare different results with details of setting and appropriate citations?

---

> > > ### Author Response · Authors · 2022-08-08
> > > **Response to Reviewer qdqB**
> > >
> > > We thank the reviewer for the feedback and definitely agree that it will be very helpful to a reader to be able to see all the different settings under which our guarantees hold within a single paper. In particular, the corollaries applying our main result to the (i) arbitrary input setting, (ii) distributional (noisy labels) setting, and (iii) distributional (input follows a distribution) setting where we care about generalization bounds. These are not apparent from our current main theorem, although they follow readily.
> > >
> > > Since the 8 page limit still applies during the rebuttal phase, and it is difficult to fit the table into the current version of the paper without modifying the rest of the paper drastically, we'll plan include this table and the surrounding discussion in the subsequent version of the paper.

---

> > > > ### Author Response · Authors · 2022-08-09
> > > > **Response to Reviewer qdqB**
> > > >
> > > > We hope that the rebuttal clarifies the questions and concerns raised by the reviewer, as well outline our approach to improving the readability of the paper upon getting the additional page availability. We would be very happy to discuss any further questions about the work and would appreciate if the reviewer might consider an appropriate increase in score if they deem that their concerns are adequately addressed.

---

### Meta-Review · Area_Chair_kx6S · 2022-08-29

**Recommendation:** Accept
**Confidence:** Certain

**Metareview:**

This submission studies the problem of active (or query) learning for linear regression. It provides label query bounds in terms of novel parameters.

All reviewers have appreciated novelty and quality of the results. The problem considered is also clearly of interest to the NeurIPS (theory) community.

**Award:**

No

---

### Decision · Program_Chairs · 2022-09-14

Accept